# Handling Tabular Data under Coupled Shifts of Feature Missingness and Distributional Change

## Abstract

Tabular data plays a vital role in a wide range of real-world applications. However, previous methods for tabular data learning primarily focused on closed environments, overlooking the fact that feature missingness and distributional shift issues can occur simultaneously in open environments. In this paper, we first investigate **R**obust **T**abular prediction under the **C**oupled **S**hifts of feature missingness and distributional change, namely *RTCS* problem. We identify three challenges in *RTCS*, where column missingness and distribution shifts are interdependent and mutually inhibitive: (1) the coexistence of column missingness and distribution shifts leads to severe performance degradation, for which no effective solutions currently exist; (2) under distribution shifts, it is inherently difficult to obtain reliable statistical patterns for imputing missing features; and (3) mitigating information loss from missing features while maintaining robustness to distribution shifts remains highly challenging. To this end, we propose **K**nowledge-**G**uided **C**oupled **S**hift handler for **Tab**ular data, namely *KGCS4Tab*, which effectively disentangles feature missingness from distribution shifts by performing column imputation by constructing Knowledge-Guided recovery rules, and adapts to unknown distributions through model selection with theoretical guarantee. Experimental results demonstrate that *KGCS4Tab* achieves a nearly 20% performance gain.

## 1 Introduction

Tabular data plays a critical role in real-world applications (Altman & Krzywinski, 2017) such as healthcare (Ching et al., 2018), finance (Hein et al., 2017), manufacturing (Hein et al., 2017) and economics (Salehpour & Samadzamini, 2024). Machine learning models (Ke et al., 2017; Arik & Pfister, 2021; Gorishniy et al., 2021; Prokhorenkova et al., 2018) have achieved remarkable success in tabular data under closed environments where important learning factors, such as data distribution and feature spaces, remain consistent. However, real-world applications of tabular data often occur in open environments, where data distribution could change (Kolesnikov, 2023) and feature columns could decrease (Cheng et al., 2025). These challenges can lead to robustness issues, manifesting as a degradation in accuracy compared to scenarios without distribution shifts or missing features.

Recently, growing research efforts have been devoted to improving the robustness for handling tabular data in open environments. To address feature missingness, researchers proposed feature-robust models such as LightGBM (Ke et al., 2017), StableMiss (Zhu et al., 2023), and DAMS (Zhou et al., 2023a), which adapt to missing patterns through information gain, feature–mask decorrelation, and relative missingness estimation, respectively. For distribution shift issue, various shift-robust approaches (Zhou et al., 2025; Ren et al., 2024; Kim et al., 2024) have been developed specifically for tabular data learning. For instance, FTAT (Zhou et al., 2025) monitors data distribution through prediction confidence and adapts to it by minimizing prediction entropy during the test phase.

These approaches typically address either feature missingness (Du et al., 2024) or distribution shifts (Wu et al., 2021; Zhou et al., 2025; 2023b) in isolation. However, these two types of shifts often occur simultaneously in practice (Fig.1). For example, in medical diagnostics (Cheng et al., 2025; Sekkarie, 2024), when models are transferred from regions with well-developed healthcare infrastructure to regions with less-developed infrastructure, differences in testing equipment can lead

Table 1: Accuracy of TabM, IRM, Masked, and their combination under open environments. Degraded performance is underlined.

| Method | ANES | Hypertension | ACSincome | ACSpubcov |
|---|---|---|---|---|
| TabM | 80.14 | 58.37 | 78.95 | 63.45 |
| IRM | 78.89 | 56.28 | 77.67 | 47.45 |
| Masked | 79.87 | 58.80 | 79.73 | 63.19 |
| Masked + IRM | 78.93 | 55.59 | 78.01 | 50.56 |

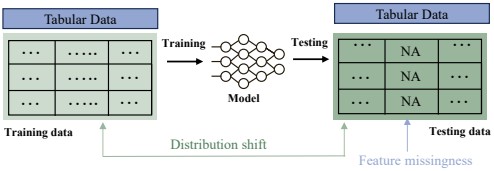

Figure 1: The example illustration of feature missingness and distribution shifts in open-environment tabular prediction.

to foreseeable feature missingness, and differences in population health status can induce distribution shifts. As shown in Table 1, standard feature missing robust method (masked training), distribution shift robust method (IRM (Arjovsky et al., 2019)) and their collaboration cannot address the coupled challenges of feature missingness and distribution shifts. Consequently, achieving robust prediction under their joint influence remains a critical yet largely underexplored problem, with significant implications for real-world applications (Altman & Krzywinski, 2017).

To this end, we study the practical problem of training tabular models to achieve robust predictions when certain column of features are *entirely* unobserved and the data distribution shifts at test time. We name this problem **R**obust **T**abular prediction under the **C**oupled **S**hifts of feature missingness and distributional change (*RTCS*). To address the *RTCS* problem, we first conduct an in-depth investigation into the *RTCS* problem. Three observations we obtained under the *RTCS* scenario provide insights into the three key challenges in designing solutions for *RTCS*: (a) The coexistence of column missingness and distribution shifts leads to severe performance degradation, neither feature missing robust methods nor distribution robust methods can address this problem; (b) It is inherently difficult to obtain reliable statistical patterns for imputing missing features under distribution shifts; (c) The information loss caused by missing features makes it hard to adapt to distribution shifts

To address these challenges, we propose **K**nowledge-**G**uided **C**oupled **S**hift handler for **Tab**ular data, namely *KGCS4Tab* methods, which is composed of two modules: *Knowledge-Guided Feature Aligner* and *Distribution-Aware Model Selector*, which effectively disentangles feature missingness from distribution shifts by performing column imputation by constructing Knowledge-Guided recovery rules, and adapts to unknown distributions through model selection with theoretical guarantee. Experimental results demonstrate the effectiveness and generalizability of our proposal, achieving a nearly 20% performance gain over state-of-the-art approaches across four datasets, and consistently improving performance across five different tabular backbone models.

To summarize, the contributions of this paper are threefold:

- We investigate the robust predictions for the tabular data under the *RTCS* problem, identifying three key challenges: the coexistence of column missingness and distribution shifts, unreliable imputation methods, and ineffective distribution robust methods.

- We propose a novel approach, *KGCS4Tab*, which consists of *Knowledge-Guided Feature Aligner* and *Distribution-Aware Model Selector* to address the challenges of columns imputation by constructing knowledge-guided recovery rules and performing data-augmented training, and adapts to unknown distributions through model selection, respectively.

- We evaluate the *KGCS4Tab* approach on four tabular benchmarks with the coexistence of column missingness and distribution shifts using 5 representative tabular models, demonstrating the effectiveness and generalizability of our method.

## 2 RELATED WORKS

**Tabular Machine Learning** Tabular Machine Learning aims to model tabular data for tasks such as classification and regression. Recently, decision-tree based models like LightGBM (Ke et al., 2017)

and CatBoost (Prokhorenkova et al., 2018) are strong and efficient baselines for tabular data. Deep learning methods for tabular such as FT-Transformer (Gorishniy et al., 2021), TabPFN (Hollmann et al., 2023a; 2025), and TabM (Gorishniy et al., 2025), achieve good performance on closed environments, and may suffer performance degradation when real-world distribution shifts occur.

**Distribution Shift Robust Methods** Existing distribution shift robust methods fall into two categories: domain generalization, domain adaptation and test-time adaptation. Domain generalization includes ERM (Vapnik, 1999) and its variants (IB_ERM (Ahuja et al., 2021), EQRM (Eastwood et al., 2022), VREx (Krueger et al., 2021), ERM++ (Teterwak et al., 2025)), as well as IRM (Arjovsky et al., 2019) and DRO (Sagawa et al., 2019). For example, IB_ERM (Ahuja et al., 2021) applies the Information Bottleneck principle to ERM for improved robustness. Domain adaptation (Kang et al., 2019; Li et al., 2020; Du et al., 2021) usually bridges the domain gap by minimizing the distribution discrepancy with partial target information. Test-time adaptation methods include TTT (Gandelsman et al., 2022), TTT++ (Liu et al., 2021), and approaches that adapt only during testing, such as AdapTable (Kim et al., 2024), TabLog (Ren et al., 2024), and FTAT (Zhou et al., 2025).

**Feature Missing Robust Methods** Recent methods for feature-missing robustness fall into two categories: imputation and adaptation. For imputation, models often integrate their own strategies, e.g., LightGBM (Ke et al., 2017) assigns missing values to the child node maximizing information gain, and TabTransformer (Huang et al., 2020) treats them as a separate category. For adaptation, techniques such as subsampling, feature boosting (Fan et al., 2010; Kanter & Veeramachaneni, 2015; Li et al., 2023), methods designed for mask shift like DAMS (Zhou et al., 2023a) and StableMiss (Zhu et al., 2023; Zhu & Jiang, 2024), and more recently LLM-based approaches like LITO (Yang et al., 2024) effectively addresses class imbalance in tabular prediction using LLMs and sampling techniques, CAAFE (Hollmann et al., 2023b) and OCTree (Nam et al., 2024), have been proposed.

# 3 PROBLEM AND ANALYSIS

In this section, we first introduce the problem setting *RTCS*, including notation and problem formulation. Then, we present three observations through empirical analysis, which not only reveal the unique challenges of tabular data but also motivate the methodological designs.

## 3.1 PROBLEM FORMULATION

We consider the tabular classification problem with input space $\mathcal{X} \in \mathbb{R}^d$, where $d$ is the number of features. Each feature may assume either a continuous or a discrete value. The label space is $\mathcal{Y} \in \{0, 1\}^K$, where $K$ is the number of classes. Consequently, we use $\mathcal{D}(X), \mathcal{D}(Y), \mathcal{D}(X, Y)$ to represent the covariate, label distribution, and joint distribution.

In the *RTCS* problem, we study a scenario where a column $c_m$, which is known in advance following previous related studies (Zhou et al., 2023a; Zhu et al., 2023), is **entirely** missing at test time and the joint distribution differs between the training and testing sets which satisfy $\mathcal{D}_{tr}(Y) \neq \mathcal{D}_{te}(Y)$ and $\mathcal{D}_{tr}(X, Y) \neq \mathcal{D}_{te}(X, Y)$, while $\mathcal{D}_{tr}(X \mid Y) = \mathcal{D}_{te}(X \mid Y)$. The goal is to train models $f : \mathcal{X} \mapsto \mathcal{Y}$ that can make robust predictions under this setting.

Our assumption reflects practical real-world applications, for example, in industrial equipment failure prediction, critical monitoring (e.g., acoustic analysis) relies on expensive sensors, but in actual deployment, only basic sensing units may be retained, which causes feature missingness during deployment. Meanwhile, differences such as the climate between two factories can lead to changes in data distribution.

## 3.2 PROBLEM ANALYSIS

We have identified three observations that suggest existing methods fail to address the *RTCS* problem effectively. Moreover, these observations underscore the primary challenges in solving the *RTCS* problem and motivate the design of *KGCS4Tab* methods.

**Observation 1: The coexistence of feature missingness and distribution shifts leads to more severe performance degradation.** Observation 1 indicates that the combined effect poses greater challenges than each factor individually. We test the performance in the situation of no shifts, only

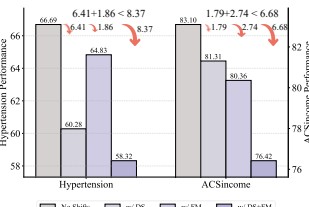 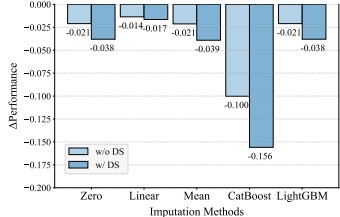 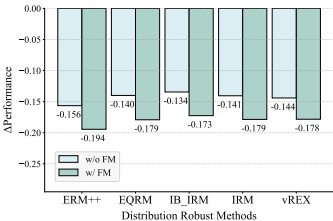

Figure 2: The performance degradation under the coexistence of feature missingness and distribution shifts. Results come from LightGBM.

Figure 3: The performance degradation under the situation of feature missingness and w/ or w/o distribution shifts (DS). Results come from LightGBM (Zero, Linear, Mean).

Figure 4: The performance degradation under the situation of distribution shifts and w/ or w/o feature missingness (FM).

distribution shifts (w/ DS), only feature missingness (w/ FM) and coupled shifts (w/ DS+FM). As shown in Fig.2, the performance all drops when any type of shifts occurs. More importantly, when distribution shifts and feature missingness occur simultaneously, their combined effect leads to a performance drop greater than the sum of their individual impacts, highlighting the significant challenge posed by the coupled problem. We provide a detail analysis on this observation about the performance degradation in Appendix C.4., highlighting the significant challenge posed by the coupled problem.

**Observation 2: Distribution shifts invalidate the statistical invariance assumption underlying existing imputation methods, making them unreliable.** Common imputation approaches, such as zero and mean imputation in deep learning (Gorishniy et al., 2021; Arik & Pfister, 2021), or model-specific strategies in gradient boosting trees (Ke et al., 2017), rely heavily on static statistical properties of the training distribution. However, once the distribution shifts, these assumptions no longer hold, and the resulting imputations become unreliable. As illustrated in Fig. 3, in ACSincome dataset, the imputation methods show a worse performance when distribution occurs, as existing imputation strategies cannot effectively compensate for the lost information. This underscores a critical gap in current practice. To achieve robustness in tabular learning, it is essential to move beyond conventional imputation and design feature imputation strategies that remain effective under distribution change—an indispensable requirement in the *RTCS* problem.

**Observation 3: The information loss induced by feature missingness undermines the effectiveness of shift robust methods.** Distribution robust approaches, such as Invariant Risk Minimization (Arjovsky et al., 2019), aim to achieve robustness by identifying invariant patterns across different domains. While such patterns are often readily identifiable in modalities like images and text due to low-level shared structures. However, in tabular settings, relationships between features and the target can vary significantly across domains, making it difficult to discover a universal invariant mechanism. Feature missingness further aggravates this problem by obscuring domain-specific dependencies and erasing critical signals, preventing distribution robust methods from reliably extracting true invariances. As illustrated in Fig. 4, in the ACSpubcov dataset, we train the model on the source domain and evaluated on the target domain w/ and w/o feature missingness, the performance of distribution robust methods deteriorates in the presence of missing features, highlighting the need for developing tailored distribution robust strategies in the *RTCS* problem.

## 4 METHOD

As discussed in analysis section, the *RTCS* problem encompasses three major challenges:

(a) The coexistence of column missingness and distribution shifts exists in tabular data, but cannot be addressed by either missing robust methods or shift robust methods.

(b) It is inherently difficult to obtain reliable statistical patterns for imputing missing features under distribution shifts.

(c) The information loss caused by missing features makes it hard to adapt to distribution shifts.

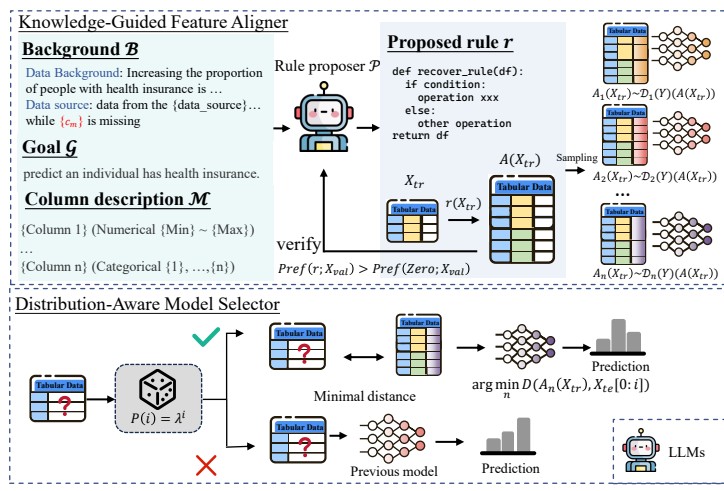

Figure 5: The overall illustation of *KGCS4Tab* approach.

Inspired by these challenges, we propose *KGCS4Tab*, a framework that tackles the *RTCS* problem by disentangling feature missingness from distribution shifts. *KGCS4Tab* consists of two components: the *Knowledge-Guided Feature Aligner* (KGFA) and the *Distribution-Aware Model Selector* (DAMS). For a batch of test data indexed by $j$, the final prediction is given by:

$$\hat{Y} = f_i(r(X_{te}[0:j]))$$
$$\text{s.t.} \quad i = \arg\min_{k\in[n]} D(A_k(X_{tr}), X_{te}[0:j]), \tag{1}$$

where $f_i$ is the model trained by KGFA on the augmented dataset $A_k$ with the ability to leverage the imputation rule $r$ under the corresponding data distribution, model $f_i$ is selected by the DAMS on the minimal dataset-level distance $D$ between the augmented training set and the test set. In summary, *KGCS4Tab* separates feature missingness from distribution shifts via two components: KGFA, which learns imputation rules and trains distribution-specific models, and DAMS, which estimates the test distribution using imputation information and selects the most suitable model based on dataset-level distance $D$. We describe each component in detail below.

## 4.1 KNOWLEDGE-GUIDED FEATURE ALIGNER

First, we aim to realize reliable imputation for the missing columns. The existing solutions (Prokhorenkova et al., 2018) fail because these methods transfer the bias of the training set to the test set. Therefore, we need an imputation method to avoid the usage of shifted information.

Motivated by empirical observations, we proposed an instance-wise feature imputation method with the guidance of dataset background knowledge (Altman & Krzywinski, 2017). For a tabular dataset, it consists of background information $\mathcal{B}$, a prediction goal $\mathcal{G}$, and column descriptions $\mathcal{M}$. As shown in Fig.5, the background information includes the data source, the underlying purpose behind the data collection, and so on. The prediction goal is the variable to be predicted, and the column descriptions are provided by the column names, reflecting the meaning of each field to help the model understand the table structure. As a large-scale knowledge base, the large language model (LLM) contains extensive background knowledge that can assist in identifying features related to missing columns. We regard the LLM as a rule proposer $\mathcal{P}$, leveraging prompt-based activation to elicit its internal knowledge and employing reasoning to generate practically meaningful rules for column-wise missing value imputation. The proposed rule $r$ to recover a missing column $c_m$ is

$$r \leftarrow \mathcal{P}(c_m; \mathcal{B}, \mathcal{G}, \mathcal{M}) \tag{2}$$

We then verify the quality of $r$ by evaluating its performance on the validation set held out from the training set using LightGBM. It is accepted if it improves upon zero imputation, indicating that it provides useful information for subsequent processing.

$$\text{Accept}(r) = \mathbf{1}[\,\text{Perf}(r; X_{val}) > \text{Perf}(\text{Zero}; X_{val})\,] \tag{3}$$

The rule $r$ from $\mathcal{P}$ uses selected sample features and mathematical operations for instance-wise imputation of $c_m$. Directly applying $r$ to the test set is infeasible due to a semantic gap, where the generated imputation rules may deviate from the original feature semantics. This issue arises because the provided information includes only the feature columns of the table, offering highly limited contextual cues. To resolve this, we first apply $r$ on the training set $X_{tr}$ to augment the data, which can be expressed as:

$$A(X_{tr}) = \text{concat}(X_{tr}, r(X_{tr})) \tag{4}$$

We leverage the imputed information in the augmented dataset along with the distributional information from different data distributions to train a series of models, enabling them to make accurate predictions under the absence of $c_m$ and its corresponding data distribution. More formally, we first sample the augmented training data on $n$ different label distributions, each of which is

$$A_i(X_{tr}) \sim \mathcal{D}_i(Y)(A(X_{tr})) \tag{5}$$

where $\mathcal{D}_i(Y)$ represent the label distribution.

Then we minimize the empirical risk on each dataset to train each model, i.e.

$$f_i = \arg\min_{f_i} \mathbb{E}_{(x,y) \sim A_i(X_{tr})}[R(f_i(x), y)] \tag{6}$$

### 4.2 Distribution-Aware Model Selector

We introduce the *Distribution-Aware Model Selector*, which fully exploits the imputation information and models to compute the data distribution and accurately predict by a dataset-level distance based distribution estimating algorithm.

Firstly, we explain how to compute dataset-level distances. In tabular data, feature columns are the fundamental units of a dataset, and the overall dataset distribution is determined by the distributions of its columns. To capture distributional differences between datasets, we first compute column-level distribution distances, with separate designs for continuous and categorical features.

For a continuous column $h$, we use Gaussian kernel density estimation (Węglarczyk, 2018) to approximate the probability density of each continuous feature and then we compute the distributional difference using the first-order Wasserstein distance.

$$d_{con}(h_{tr}, h_{te}) = W_1(h_{tr}, h_{te}) = \inf_{\gamma \in \Gamma(\hat{h}_{tr}, \hat{h}_{te})} \mathbb{E}_{(x,y) \sim \gamma}[\|x - y\|] \tag{7}$$

where $\hat{h}_{tr}, \hat{h}_{te}$ respectively represent the probability density function obtained by Gaussian kernel density estimation for column $h$ in training and testing set, $\Gamma$ denotes the joint distributions of $\hat{h}_{tr}$ and $\hat{h}_{te}$.

For a discrete column $k$, we compute the distance based on Kullback–Leibler divergence

$$d_{dis}(k_{tr}, k_{te}) = \sqrt{2D_{\text{KL}}(k_{tr} \parallel k_{te})} = \sqrt{2\sum_{i=1}^{m} k_{tr}(i) \log\left(\frac{k_{tr}(i)}{k_{te}(i)}\right)} \tag{8}$$

where $m$ represents the number of possible values for column $k$.

The overall difference between datasets is then computed by

$$D(X_{tr}, X_{te}) = \frac{1}{p+q}\left(\sum_{i=1}^{p} d_{dis}(k_{tr}^i, k_{te}^i) + \sum_{j=1}^{q} d_{con}(h_{tr}^j, h_{te}^j)\right) \tag{9}$$

$p, q$ denote the number of discrete features and continuous features, $k^i$ and $h^j$ represent the i-th discrete column and j-th continuous column, respectively.

Secondly, we adopt a probabilistic selection strategy to decide whether to compute the distributional distance between the data in batch 0 through batch $j$ and the source training data at the $j$-th batch,

thereby reducing computational burden. This approach enables the dynamic selection of the most suitable model for subsequent predictions based on distributional alignment. For batch index $j$, the probability of selecting a model is

$$P(i) = \lambda^j, \lambda \in [0, 1] \tag{10}$$

We select the model that was trained on the augmented training set with the closest distance to $0 - j$ batch data, and use it for subsequent predictions until the next model selection stage. We use $X_{te}[0:j]$ to represent the $0 - j$ batch data in the test set. The selected model $f_i$ used for prediction satisfies

$$i = \arg \min_{k \in [n]} D(A_k(X_{tr}), X_{te}[0:j]) \tag{11}$$

### 4.3 THEORICAL ANALYSIS

In this section, we provide a theoretical justification for the distance computation method and the probabilistic selection strategy.

**Theorem 1 (T1-Talagrand inequality (Bobkov & Götze, 1999; Talagrand, 1996))** For a distribution $\mu$ consists of a finite number of Gaussian kernels, then for any probability distributions $\mu$ and $\nu$, we have $W_1(\nu, \mu) \leq \sqrt{2C \cdot D_{\mathrm{KL}}(\nu \| \mu)}$

**Remark 1** Theorem 1 shows that distance computation for continuous features shares the same theoretical upper bound as that for discrete features. Consequently, in our method, the distance measures for discrete and continuous features are unified on the same scale, ensuring their comparability and equal importance in the overall distance computation.

For the probabilistic selection strategy, we derive an upper bound on the expected error between the distribution estimated using our strategy and the distribution estimated from the entire dataset. We first define some terms. Let the data in batch $i$ be the currently observed data. We denote the estimated distribution of the $j$-th feature column from batches 0 to $i$ as $c_{0:i}^j$, and the corresponding distribution estimated from the full test dataset as $c_{0:N}^j$. The maximum distribution estimation error is defined as $\Delta = \max_{i,j} \| c_{0:i}^j - c_{0:N}^j \|$. The expected difference between the error under this strategy and the error using the full dataset is given by $\mathbb{E}[|D(X_{tr}, X_{te}[0:i]) - D(X_{tr}, X_{te})|]$. Considering the probabilistic selection strategy described in Section 4.2, we provide an upper bound on this expected difference.

**Theorem 2** For the probabilistic selection strategy with $P(\lambda) = \lambda^i$, the expected gap $\mathbb{E}[|D(X_{tr}, X_{te}[0:i]) - D(X_{tr}, X_{te})|]$ is bounded by $\mathcal{O}\left(p\sqrt{2C\Delta} + qL\Delta\right)$ where $C$, $L$ are constants and $p$ and $q$ denote the numbers of discrete and continuous features, respectively.

**Remark 2** Theorem 2 demonstrates that our probabilistic strategy effectively balances computational efficiency and accuracy. The derived upper bound guarantees that the estimation error, relative to using the full dataset, remains small and is primarily determined by the dataset's intrinsic properties. By probabilistically deciding whether to compute distances for each batch, our method avoids expensive exhaustive calculations.

**Remark 3** Theorem 1 ensures scale consistency across different feature types, while Theorem 2 provides an error upper bound for the probabilistic selection strategy. Theorem 1 ensures accurate computation of distribution distances, and the ($\Delta$) term in Theorem 2 reflects the effect of imputation as the training set is augmented via imputation during training, thus encoding the imputation rules. So, the error bound naturally includes the additional error from imputation, demonstrating that our method is robust to imputation while remaining computationally efficient. We provide the performance upper bound as well as the performance recovery ratio under our strategy in the Appendix D.2.

## 5 EXPERIMENTS

In this section, we conduct experiments to answer the following research questions:

RQ 1: Can *KGCS4Tab* maintain robust prediction performance when facing feature missingness and distribution shifts?

RQ 2: Does the rule proposer $\mathcal{P}$ in *KGCS4Tab* lead to better distribution estimation?

RQ 3: Can *KGCS4Tab* maintain its robustness when encountering more complex scenarios?

To answer those research questions, We train models on the training set, select the best by validation, and test with a missing column (chosen by GBDT importance) and distribution shifts in TABLESHIFT benchmarks. We evaluate with accuracy (ACC) and F1 (mean±std), comparing with shift-robust methods, missing-robust methods, and tabular LLMs across five backbones. The detailed experiment setup shown in Appendix C.2

## 5.1 EMPIRICAL RESULTS

**RQ 1:** Can *KGCS4Tab* maintain robust prediction performance when facing feature missingness and distribution shifts?

Table 2: Performance of *KGCS4Tab* approach and comparison methods on 4 datasets with 5 tabular backbones, the best performance is in **bold**

| Methods | | ANES | | Hypertension | | ACSincome | | ACSpubcov | |
|---|---|---|---|---|---|---|---|---|---|
| | | ACC | F1 | ACC | F1 | ACC | F1 | ACC | F1 |
| **Fundamental Tabular Model** | | | | | | | | | |
| GBDT | | 79.13 ± 0.00 | 84.32 ± 0.00 | 56.82 ± 0.00 | 50.39 ± 0.00 | 74.44 ± 0.00 | 57.57 ± 0.00 | 54.40 ± 0.00 | 46.91 ± 0.00 |
| LightGBM | | 79.87 ± 0.00 | 84.61 ± 0.00 | 56.48 ± 0.00 | 49.09 ± 0.00 | 76.74 ± 0.00 | 64.12 ± 0.00 | 64.68 ± 0.00 | 65.64 ± 0.00 |
| CatBoost | | 79.86 ± 0.00 | 84.60 ± 0.00 | 58.32 ± 0.00 | 54.98 ± 0.00 | 76.42 ± 0.00 | 62.93 ± 0.00 | 67.14 ± 0.00 | 69.42 ± 0.00 |
| MLP | | 78.67 ± 0.44 | 83.86 ± 0.14 | 55.95 ± 0.91 | 48.70 ± 2.93 | 77.56 ± 0.29 | 67.46 ± 0.96 | 49.41 ± 0.24 | 38.05 ± 0.44 |
| TabM | | 80.14 ± 0.07 | **84.69 ± 0.04** | 58.37 ± 0.40 | 55.05 ± 1.27 | 78.95 ± 0.07 | 69.98 ± 0.11 | 63.45 ± 0.91 | 63.05 ± 1.84 |
| **Shift Robust Methods** | | | | | | | | | |
| ERM | | 79.13 ± 0.30 | 83.97 ± 0.05 | 56.87 ± 0.50 | 51.75 ± 1.66 | 77.91 ± 0.11 | 68.55 ± 0.22 | 48.10 ± 0.77 | 34.93 ± 1.96 |
| IRM | | 78.89 ± 0.30 | 83.85 ± 0.09 | 56.28 ± 0.24 | 49.73 ± 0.76 | 77.67 ± 0.07 | 67.87 ± 0.33 | 47.45 ± 0.77 | 33.32 ± 1.93 |
| vREX | | 79.37 ± 0.28 | 84.08 ± 0.06 | 56.35 ± 0.37 | 50.01 ± 1.23 | 77.38 ± 0.27 | 66.96 ± 0.70 | 47.43 ± 0.46 | 33.22 ± 1.22 |
| IB_IRM | | 79.30 ± 0.38 | 84.14 ± 0.13 | 56.45 ± 1.09 | 50.30 ± 3.56 | 77.45 ± 0.12 | 67.22 ± 0.30 | 47.47 ± 0.24 | 33.18 ± 0.67 |
| IB_ERM | | 79.16 ± 0.13 | 84.07 ± 0.09 | 56.25 ± 0.18 | 49.65 ± 0.63 | 77.85 ± 0.19 | 68.35 ± 0.47 | 47.18 ± 0.52 | 32.45 ± 1.40 |
| EQRM | | 78.22 ± 0.21 | 83.54 ± 0.12 | 56.40 ± 1.44 | 50.21 ± 4.79 | 77.83 ± 0.09 | 68.39 ± 0.30 | 46.53 ± 0.29 | 30.77 ± 0.72 |
| GroupDRO | | 79.09 ± 0.35 | 84.06 ± 0.12 | 51.41 ± 7.01 | 33.34 ± 23.75 | 77.83 ± 0.04 | 68.27 ± 0.17 | 47.48 ± 1.05 | 33.16 ± 2.63 |
| ERM++ | | 79.17 ± 0.54 | 84.00 ± 0.07 | 45.86 ± 6.06 | 14.60 ± 20.64 | 77.41 ± 0.68 | 67.05 ± 2.17 | 44.68 ± 1.51 | 25.79 ± 4.32 |
| MLP | TTT++ | 72.46 ± 0.81 | 80.87 ± 0.38 | 58.69 ± 0.86 | 66.17 ± 6.75 | 78.78 ± 0.58 | 70.35 ± 2.11 | 39.28 ± 0.65 | 11.02 ± 2.23 |
| | MT3 | 79.24 ± 0.26 | 84.30 ± 0.14 | 41.92 ± 0.30 | 1.54 ± 1.24 | 78.91 ± 0.08 | 70.49 ± 0.35 | 41.63 ± 1.30 | 17.11 ± 3.84 |
| TabM | TTT++ | 72.18 ± 1.34 | 80.72 ± 0.63 | 54.59 ± 1.30 | 43.96 ± 3.54 | 73.67 ± 1.60 | 55.56 ± 4.70 | 37.02 ± 0.31 | 2.17 ± 1.00 |
| | MT3 | 72.65 ± 8.13 | 80.43 ± 4.79 | 54.20 ± 5.04 | 59.66 ± 18.42 | 77.23 ± 0.24 | 64.91 ± 0.78 | 38.14 ± 0.32 | 5.97 ± 1.07 |
| **Missing Robust Methods** | | | | | | | | | |
| DAMS-adjust | | 70.13 ± 0.00 | 61.22 ± 0.00 | 58.43 ± 0.00 | 36.88 ± 0.00 | 48.83 ± 0.00 | 43.57 ± 0.00 | 59.80 ± 0.00 | 55.39 ± 0.00 |
| StableMiss | | 59.44 ± 0.00 | 37.28 ± 0.00 | 41.57 ± 0.00 | 29.36 ± 0.00 | 60.20 ± 0.00 | 37.58 ± 0.00 | 36.37 ± 0.00 | 26.67 ± 0.00 |
| **Tabular Large Language Models** | | | | | | | | | |
| TabLLM | | 78.07 ± 0.00 | 81.65 ± 0.00 | 52.98 ± 0.00 | 53.09 ± 0.00 | 72.15 ± 0.00 | 57.44 ± 0.00 | 64.54 ± 0.00 | 69.49 ± 0.00 |
| TableGPT2 | | 50.18 ± 0.00 | 65.86 ± 0.00 | 53.49 ± 0.00 | 61.77 ± 0.00 | 53.51 ± 0.00 | 56.04 ± 0.00 | 69.67 ± 0.00 | 68.73 ± 0.00 |
| **Ours** | | | | | | | | | |
| GBDT+Ours | | 80.01 ± 0.10(+0.88) | 84.50 ± 0.06(+0.18) | 63.40 ± 0.05(+6.58) | 69.37 ± 2.62(+18.98) | 80.06 ± 0.44(+5.62) | 74.69 ± 0.62(+16.52) | 70.36 ± 1.03(+15.96) | 75.45 ± 1.90(+28.54) |
| LightGBM+Ours | | 80.38 ± 0.09(+0.51) | 84.60 ± 0.07(-0.01) | **63.75 ± 0.10**(+7.27) | 69.96 ± 2.24(+20.87) | 80.86 ± 0.57(+4.12) | 75.42 ± 0.52(+11.30) | 73.59 ± 0.35(+8.91) | 78.83 ± 0.54(+13.19) |
| CatBoost+Ours | | **80.41 ± 0.15**(+0.55) | 84.56 ± 0.11(-0.04) | 63.46 ± 0.01(+5.14) | 69.53 ± 2.34(+14.55) | 81.21 ± 0.61(+4.79) | **76.07 ± 0.54**(+13.14) | **74.55 ± 0.42**(+7.41) | **79.69 ± 0.30**(+10.27) |
| MLP+Ours | | 79.21 ± 0.26(+0.54) | 84.02 ± 0.10(+0.16) | 59.70 ± 1.00(+3.75) | 59.92 ± 2.81(+11.22) | 79.70 ± 0.51(+2.14) | 73.11 ± 0.47(+5.65) | 58.71 ± 6.10(+9.30) | 55.90 ± 11.84(+17.85) |
| TabM+Ours | | 80.36 ± 0.04(+0.22) | 84.68 ± 0.12(-0.01) | 61.67 ± 2.32(+3.3) | **71.95 ± 1.30**(+16.9) | **81.26 ± 0.60**(+2.31) | 75.94 ± 0.77(+5.96) | 72.81 ± 0.16(+9.36) | 77.49 ± 0.35(+14.44) |

We present detailed experimental results in Table 2 to address this question. The results show that our *KGCS4Tab* approach outperforms existing shift-robust methods, missing-robust methods, and tabular large language models in major cases, while achieving competitive performance in the remaining cases, demonstrating its effectiveness across various tabular datasets and backbone models. Note that the performance gain on ANES is relatively small due to lower feature discriminative power: information gain for training and test sets (with distribution shift and missing features) shows a decrease of 0.012 for the ANES test set (0.079→0.067), whereas Hypertension increased by 0.027 (0.033→0.060). Moreover, *KGCS4Tab* consistently improves performance across five different backbone types, highlighting its versatility and robustness, which significantly enhances its practical utility for tabular prediction.

**RQ 2:** Does the rule proposer $\mathcal{P}$ in *KGCS4Tab* lead to better distribution estimation?

To evaluate the effectiveness of the rule proposer $\mathcal{P}$ in KGFA, we replaced it with zero and mean imputation, conducting experiments on ANES and Hypertension datasets using a LightGBM backbone. KL divergence was used to measure the distance between the true label distribution and its estimate. As shown in Table 3, $\mathcal{P}$ achieves more accurate distribution estimation than conventional imputation, improving predictive performance (Fig. 6). The recovery rules proposed by our method provide more useful information for subsequent processing, thereby facilitating distribution estimation. As further analyzed in the appendix D.1, the quality of these rules leads to measurable performance improvements. These results indicate that our imputation strategy not only ensures reliable distribution estimation but also synergizes with subsequent training and modeling, forming an integrated framework that jointly addresses feature missingness and distribution shift.

Table 3: Comparison of estimated label distribution of $\mathcal{P}$ and other imputation methods

| Imputation | ANES | Hypertension |
|---|---|---|
| Zero | 0.0085 | 0.2320 |
| Mean | 0.0034 | 0.2320 |
| Ours | **0.0016** | **0.0080** |

Additionally, we conducted experiments comparing our method with the classical impute-then-train paradigm and the approach that directly applies the proposed rules, in order to validate the effectiveness of the rule proposer. As shown in the table 4, our method outperforms the classical impute-then-train approach and also achieves better performance than directly applying the proposed rules. This indicates that our method can better model missing columns under coupled variations through the proposed rules. Meanwhile, the data augmentation training process effectively bridges the potential "semantic gap."

Table 4: Accuracy comparison with direct imputation and imputation-training methods

| Method | ANES | Hypertension |
|---|---|---|
| LightGBM | 79.87 | 56.48 |
| LightGBM+direct | 79.80 | 56.68 |
| MICE | 79.85 | 59.73 |
| LR | 79.78 | 59.57 |
| LightGBM+Ours | **80.38** | **63.75** |

**RQ 3:** Can *KGCS4Tab* maintain its robustness when encountering more complex scenarios?

To further assess the generalization capability and robustness of the *KGCS4Tab* method under more complex scenarios, we conducted experiments with 2–3 missing columns on 2 datasets. In these settings, KGFA was applied to construct recovery rules for each missing column. In Table 9, our method consistently sustained stable performance under multi-column missingness. These results not only verify the effectiveness of our approach in complex scenarios but also underscore its strong robustness and adaptability. Detailed results are included in our Appendix C.5 due to space limits.

Table 5: Performance of *KGCS4Tab* approach and comparison methods (with best performance) under 2 / 3 columns missing

| Methods | ANES | | Hypertension | |
|---|---|---|---|---|
| | ACC | F1 | ACC | F1 |
| LightGBM | 78.36/76.82 | 83.66/82.96 | 46.94/45.19 | 23.25/16.03 |
| Shift Robust Methods | 79.93/78.54 | 84.48/83.84 | 56.14/54.33 | 58.47/55.10 |
| Missing Robust Methods | 60.76/63.03 | 40.76/46.34 | 58.43/58.42 | 36.88/36.88 |
| Tabular LLMs | 48.49/47.88 | 49.34/46.47 | 52.45/51.61 | 58.53/61.81 |
| LightGBM+Ours | **80.33/80.22** | **84.60/84.55** | **61.10/60.60** | **66.23/64.94** |

Additionally, we conduct experiments to further evaluate the effectiveness of *KGCS4Tab* in more complex settings involving partial missingness. As shown in Table 6, across a range of missing ratios, our method consistently achieves performance gains over the baselines. These results demonstrate that *KGCS4Tab* remains effective even when only a portion of each column is observed, highlighting the robustness and general applicability of our approach.

Table 6: Performance of *KGCS4Tab* in partial missing scenarios

| Missing ratio | LightGBM | | LightGBM+Ours | |
|---|---|---|---|---|
| | ANES | Hypertension | ANES | Hypertension |
| 0.2 | 79.89 | 57.25 | **80.30** | **63.88** |
| 0.4 | 79.87 | 57.24 | **80.31** | **63.89** |
| 0.6 | 79.89 | 57.26 | **80.30** | **63.88** |
| 0.8 | 79.88 | 57.27 | **80.28** | **63.92** |

## 5.2 FURTHER ANALYSIS

**Ablation Study.** We assess the effectiveness of the KGFA and DAMS modules using the LightGBM. In Table 7, we present the average performance of our method with the KGFA and DAMS ablated. Using either KGFA or DAMS alone leads to performance improvements over the baseline LightGBM, which validates the ef-

Table 7: Ablation study of our method, experiments are conducted with LightGBM. Best performance is in **bold**

| Method | ANES | | Hypertension | |
|---|---|---|---|---|
| | ACC | F1 | ACC | F1 |
| LightGBM | 79.87 ±0.00 | 84.61 ±0.00 | 56.48 ±0.00 | 49.09 ±0.00 |
| Ours w/o DAMS | 79.93 ± 0.05 | 84.61 ± 0.02 | 59.93 ± 0.32 | 57.87 ± 0.51 |
| Ours w/o KGFA | 80.32 ± 0.04 | 84.52 ± 0.10 | 60.58 ± 1.30 | 62.81 ± 5.28 |
| Ours+LightGBM | **80.38 ± 0.09** | **84.59 ± 0.07** | **63.75 ± 0.10** | **69.96 ± 2.24** |

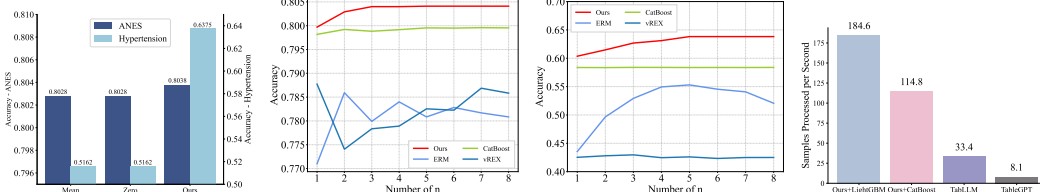

Figure 6: Rule proposer vs. other imputation methods

Figure 7: Performance on ANES with varying model counts

Figure 8: Performance on Hypertension with varying model counts

Figure 9: Comparison of running time between Ours and tabular LLMs

fectiveness of KGFA and DAMS. The complete *KGCS4Tab* method achieves the best performance, demonstrating that the KGFA and DAMS work jointly to address the challenges posed by *RTCS*, thereby validating their effectiveness.

**Effect of Number of Source Models.** To examine the effect of the number of source models in *KGCS4Tab*, we conducted experiments using CatBoost with varying numbers of source models $n$, and compared our method with ensemble versions of domain generalization baselines. The performance of our method steadily improves with more source models and eventually stabilizes, whereas the ensemble baselines show no such trend (Fig.7 and 8). The result suggests that once the augmented dataset covers the test distribution, the model predicts reliably, indicating performance gains arise from the design and interplay of our method's two modules, rather than using multiple models. We also provide the performance comprises with direct ensembling our trained model in KGFA in Appendix C.7

**Adaptation to Different Model Structure** To validate whether our method is compatible with large language models of different architectures, we use the same set of prompts and employ Qwen3-Max to generate the recovery rules, while keeping the remaining procedure unchanged. The experimental results are shown in the table. As observed, the performance of the Qwen series models is comparable to that of DeepSeek-V3 used in our main experiments. These results indicate that our proposed framework is compatible with multiple language model architectures.

Table 8: Accuracy of our approach using Qwen3-max

| Methods | ANES | Hypertension |
|---|---|---|
| LightGBM | 79.87 | 56.47 |
| LightGBM+Ours | 80.29 | 62.99 |
| CatBoost | 79.86 | 58.32 |
| CatBoost+Ours | **80.41** | **62.58** |

**Performance on complex tasks** To evaluate the capability of our method on multi-class classification and regression tasks, we conduct experiments on the larger-scale datasets of TableFSBench. We use accuracy (ACC) to measure the performance of multi-class classification and root mean squared error (RMSE) for regression. The experimental results are presented in the table. Our method achieves substantial improvements over the base models on these tasks and outperforms several existing baselines, demonstrating the effectiveness of the proposed approach.

Table 9: Performance of *KGCS4Tab* approach in multi-class classification and regression tasks

| Methods | eyemovements | jannis | concrete |
|---|---|---|---|
| | ACC | ACC | RMSE |
| CatBoost | 64.35 | 69.04 | 24.95 |
| LightGBM | 64.21 | 68.48 | 21.38 |
| Best Shift Robust Methods | 45.15 | 61.13 | 15.88 |
| CatBoost+Ours | **65.21** | **70.95** | **12.30** |
| LightGBM+Ours | **68.00** | **70.61** | **12.73** |

**Running Time.** We compare the running time with existing tabular large language models. As shown in Fig. 9, our method can process more samples per second. Our method consistently achieves a processing speed of over 100 samples per second, which is acceptable for real-world applications.

**Hyperparameter Robustness.** We show the robustness of hyperparameter $\lambda$ in Appendix C.6

# 6 CONCLUSION

We study **R**obust **T**abular prediction under **C**oupled **S**hifts (*RTCS*), addressing challenges from feature missingness and distributional change. We propose *KGCS4Tab* a framework addressing *RTCS* effectively. Experiments on 4 datasets show *KGCS4Tab* outperforms prior methods. Limitations are *KGCS4Tab* requires training multiple models, increasing computational cost, and complicating application to anonymized data; this can be alleviated via parallel training and using statistical guidance for imputation.

## REPRODUCIBILITY STATEMENT

For theoretical results, we give the proof process in Appendix A and B.

For the experiment results, We provide the data, code, and instructions to reproduce the main experimental results in the supplemental material. We provide all the necessary details to reproduce the main experimental results in the paper and the Appendix.

## ETHICS STATEMENT

Our work adheres to the ethical guidelines outlined by ICLR. The research presented in this paper focuses on methodological development and does not involve human subjects, sensitive personal data, or animals. All datasets used are publicly available and have been cited appropriately.

We acknowledge that any machine learning model can potentially be misused. While our method aims to improve robustness and fairness in the context of missing features and distribution shifts, it could be applied in ways that may unintentionally affect downstream decision-making systems. We encourage responsible use of the proposed framework and recommend that users consider fairness, privacy, and societal impact when deploying models trained using our approach.

We have also ensured transparency by providing sufficient experimental details, and our code and instructions for reproducing the results will be released publicly, supporting the principles of reproducibility and accountability.

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

## A  PROOF OF THEOREM 1

**Theorem 1 (T1-Talagrand inequality (Bobkov & Götze, 1999; Talagrand, 1996))** For a distribution $\mu$ consists of a finite number of Gaussian kernels, then for any probability distributions $\mu$ and $\nu$, we have $W_1(\nu, \mu) \leq \sqrt{2C \cdot D_{\mathrm{KL}}(\nu\|\mu)}$

**Proof**: For a single Gaussian Distribution $\mu_i = \mathcal{N}(m_i, \Sigma_i)$ satisfy T1-Talagrand inequality, there exists a constant $C_i$such that for any probability measure $\nu$, we have:

$$W_1(\nu, \mu_i) \leq \sqrt{2C_i D_{\mathrm{KL}}(\nu\|\mu_i)}. \tag{12}$$

For a finite number of Gaussian kernels, the mixture measure $\mu = \sum_{i=1}^n w_i \mu_i$, construct the coupling $\pi = \sum_{i=1}^n w_i \pi_i$, where $\pi_i$ is the optimal coupling between $\nu$ and $\mu_i$. According to the convexity inequality, we have:

$$W_1(\nu, \mu) \leq \sum_{i=1}^n w_i W_1(\nu, \mu_i). \tag{13}$$

According to 12, we have:

$$W_1(\nu, \mu) \leq \sum_{i=1}^n w_i \sqrt{2C_i D_{\mathrm{KL}}(\nu\|\mu_i)}. \tag{14}$$

According to the Cauchy-Schwarz Inequality, we have:

$$\sum_{i=1}^n w_i \sqrt{D_{\mathrm{KL}}(\nu\|\mu_i)} \leq \sqrt{\sum_{i=1}^n w_i} \cdot \sqrt{\sum_{i=1}^n w_i D_{\mathrm{KL}}(\nu\|\mu_i)} = \sqrt{\sum_{i=1}^n w_i D_{\mathrm{KL}}(\nu\|\mu_i)}. \tag{15}$$

For entropy, we have the convexity of relative entropy:

$$\sum_{i=1}^n w_i D_{\mathrm{KL}}(\nu\|\mu_i) \geq D_{\mathrm{KL}}(\nu\|\mu), \tag{16}$$

Combining 15 and 16, define $C = \max_i C_i$, we have:

$$W_1(\nu, \mu) \leq \sqrt{2C} \cdot \sqrt{D_{\mathrm{KL}}(\nu\|\mu)}. \tag{17}$$

Thus, the Gaussian mixture measure $\mu$ satisfies the T1-Talagrand inequality with constant $C = \max_i C_i$, i.e.,

$$W_1(\nu, \mu) \leq \sqrt{2C D_{\mathrm{KL}}(\nu\|\mu)}.$$

Therefore, the conclusion holds: For a distribution $\mu$ consists of a finite number of Gaussian kernels, then for any probability distributions $\mu$ and $\nu$, we have $W_1(\nu, \mu) \leq \sqrt{2C \cdot D_{\mathrm{KL}}(\nu\|\mu)}$. $\square$

## B  PROOF OF THEOREM 2

**Theorem 2** Suppose at batch $i$, the probability of selecting the batch for estimating the distance between training sets is given by $P(i) = \lambda^i$, where $\lambda \in [0, 1]$. Then, under this probabilistic selection strategy, the expected gap between the estimation error using the selected batches and the error using the full dataset admits the following upper bound $\mathcal{O}\left(p\sqrt{2C\Delta} + qL\Delta\right)$ where $C$, $L$ are constants and $p$ and $q$ denote the numbers of discrete and continuous features, respectively.

**Proof:**

**Lemma 1** We define $X_{te}[0:i]$ as the test data from batch 0 to batch $i$, the distance estimation as $D(X_{tr}, X_{te}[0:i])$, so the distance estimated difference between the whole data and top $i$ batch data i.e. $\|D(X_{tr}, X_{te}[0:i]) - D(X_{tr}, X_{te})\|$ is bounded with $\mathcal{O}(p\sqrt{2C\Delta} + qL\Delta)$

**Proof of Lemma 1** Consider the definition, the distance is computed with KL divergence and Wasserstein distance, which are Lipschitz continuous, we have:

$$
|D(X_{tr}, X_{te}[0:i]) - D(X_{tr}, X_{te})| = \frac{1}{p+q} |\sum_{j=1}^{q} d_{con}(h_{tr}^j, h_{0:i}^j) + \sum_{j=1}^{p} d_{dis}(k_{tr}^j, k_{0:i}^j)
$$
$$
- \sum_{j=1}^{q} d_{con}(h_{tr}^j, h_{te}^j) - \sum_{j=1}^{p} d_{dis}(k_{tr}^j, k_{te}^j)|
$$

(18)

using Pinsker inequality and Lipschitz continuity, we have:

$$
\left| d_{dis}(k_{tr}^j, k_{0:i}^j) - d_{dis}(k_{tr}^j, k_{te}^j) \right| = \left| \sqrt{2D_{KL}(k_{tr}^j \| k_{0:i}^j)} - \sqrt{2D_{KL}(k_{tr}^j \| k_{te}^j)} \right|
$$
$$
\leq \left| \sqrt{2C_1 \| k_{tr}^j - k_{0:i}^j \|} - \sqrt{2C_2 \| k_{tr}^j - k_{te}^j \|} \right|
$$

(19)

$$
\leq \sqrt{2 \max(C_1, C_2) \left| k_{0:i}^j - k_{te}^j \right|}
$$

$$
|d_{con}(h_{tr}^j, h_{0:i}^j) - d_{con}(h_{tr}^j, h_{te}^j)| \leq L|h_{0:i}^j - h_{te}^j|
$$

(20)

Combining the above formula, we have:

$$
|D(X_{tr}, X_{te}[0:i]) - D(X_{tr}, X_{te})| \leq \frac{1}{p+q} \left| \sum_{j=1}^{p} \sqrt{2 \max(C_1, C_2) \left| k_{0:i}^j - k_{te}^j \right|} + \sum_{j=1}^{q} L|h_{0:i}^j - h_{te}^j| \right|
$$

(21)

We define $\Delta = \max_{i,j} \|c_{0:i}^j - c_{0:N}^j\|, c \in \{h, k\}$, $C = \max(C_1, C_2)$, which is an inherent property of the dataset. Then we have:

$$
|D(X_{tr}, X_{te}[0:i]) - D(X_{tr}, X_{te})| \leq \frac{1}{p+q} \| \sum_{j=1}^{p} \sqrt{2C\Delta} + \sum_{j=1}^{q} L\Delta \| = \frac{1}{p+q}(p\sqrt{2C\Delta} + qL\Delta)
$$

(22)

Lemma 1 proof ends $\square$

**Proof of Theorem 2** Consider the whole process, batch $i$ is finally chosen, has a probability:

$$
\lambda^i \prod_{j=i+1}^{T} (1 - \lambda^j)
$$

(23)

So, the expectation of estimation error is

$$
\mathbb{E}[|D(X_{tr}, X_{te}[0:i]) - D(X_{tr}, X_{te})|] = \sum_{i=1}^{T} \lambda^i \prod_{j=i+1}^{T} (1 - \lambda^j)|D(X_{tr}, X_{te}[0:i]) - D(X_{tr}, X_{te})|
$$

(24)

Using Lemma 1, we have:

$$
\mathbb{E}[|D(X_{tr}, X_{te}[0:i]) - D(X_{tr}, X_{te})|] \leq \sum_{i=1}^{T} \lambda^i \prod_{j=i+1}^{T} (1 - \lambda^j)(\frac{1}{p+q}(p\sqrt{2C\Delta} + qL\Delta))
$$

(25)

Consider $\frac{1}{p+q}(p\sqrt{2C\Delta} + qL\Delta)$ is a constant, we just need to solve $\sum_{i=1}^{T} \lambda^i \prod_{j=i+1}^{T} (1 - \lambda^j)$, we solve as follow:

first we process $\prod_{j=i+1}^{T} (1 - \lambda^j)$

$$
\prod_{j=i+1}^{T} (1 - \lambda^j) = \frac{\prod_{j=1}^{T} (1 - \lambda^j)}{\prod_{j=1}^{i} (1 - \lambda^j)}
$$

Then the array can be written as:

$$\prod_{j=1}^{T}(1-\lambda^j)\sum_{i=1}^{T}\frac{\lambda^i}{\prod_{j=1}^{i}(1-\lambda^j)}$$

We compute $\sum_{i=1}^{T}\frac{\lambda^i}{\prod_{j=1}^{i}(1-\lambda^j)}$ by telescoping sum, we define $a_i = \frac{1}{\prod_{j=1}^{i-1}(1-\lambda^j)}$, we have

$$a_{i+1}-a_i = \frac{1}{\prod_{j=1}^{i}(1-\lambda^j)} - \frac{1}{\prod_{j=1}^{i-1}(1-\lambda^j)} = \frac{1}{\prod_{j=1}^{i-1}(1-\lambda^j)}\left(\frac{1}{1-\lambda^i}-1\right)$$

we can get

$$a_{i+1}-a_i = \frac{\lambda^i}{\prod_{j=1}^{i}(1-\lambda^j)}$$

Then the original sum can be written as,

$$\sum_{i=1}^{T}(a_{i+1}-a_i) = a_{T+1}-a_1$$

define $a_1 = 1$, we have:

$$\sum_{i=1}^{T}\frac{\lambda^i}{\prod_{j=1}^{i}(1-\lambda^j)} = a_{T+1}-a_1 = \frac{1}{\prod_{j=1}^{T}(1-\lambda^j)} - 1$$

So,

$$\sum_{i=1}^{T}\lambda^i\prod_{j=i+1}^{T}(1-\lambda^j) = \prod_{j=1}^{T}(1-\lambda^j)\sum_{i=1}^{T}\frac{\lambda^i}{\prod_{j=1}^{i}(1-\lambda^j)} = \prod_{j=1}^{T}(1-\lambda^j)(\frac{1}{\prod_{j=1}^{T}(1-\lambda^j)}-1)$$

We have:

$$\sum_{i=1}^{T}\lambda^i\prod_{j=i+1}^{T}(1-\lambda^j) = 1 - \prod_{j=1}^{T}(1-\lambda^j) \tag{26}$$

So, the expectation is

$$\mathbb{E}[|D(X_{tr}, X_{te}[0:i]) - D(X_{tr}, X_{te})|] \leq \sum_{i=1}^{T}\lambda^i\prod_{j=i+1}^{T}(1-\lambda^j)(\frac{1}{p+q}(p\sqrt{2C\Delta}+qL\Delta))$$
$$= \frac{1}{p+q}(p\sqrt{2C\Delta}+qL\Delta)(1-\prod_{j=1}^{T}(1-\lambda^j)) \tag{27}$$

Consider $(1-\prod_{j=1}^{T}(1-\lambda^j))$ is a constant and $T$ is determined by the testing set, which is also a constant, we have

$$\mathbb{E}[|D(X_{tr}, X_{te}[0:i]) - D(X_{tr}, X_{te})|] \leq \mathcal{O}(p\sqrt{2C\Delta}+qL\Delta) \tag{28}$$

Theorem 2 Proof ends $\square$.

# C  DETAILED EXPERIMENTAL SETUP

## C.1  DETAILS OF COMPARISON METHODS

We conduct experiments to compare *KGCS4Tab* with three kinds of shift-robust methods.

For domain generalization methods, we compare with

(1) ERM (Vapnik, 1999) (Empirical Risk Minimization) is a fundamental learning principle that seeks to minimize the average loss over the training data as a proxy for minimizing the true expected risk.

(2) IRM (Arjovsky et al., 2019) (Invariant risk minimization) enforces the optimal classifier on top of the representation space to be the same across all domains.

(3) IB_ERM (Ahuja et al., 2021) incorporates the Information Bottleneck (IB) principle into Empirical Risk Minimization (ERM)

(4) IB_IRM (Ahuja et al., 2021) incorporates the Information Bottleneck (IB) principle into Invariant risk minimization

(5) EQRM (Eastwood et al., 2022) introduces Quantile Risk Minimization based on ERM.

(6) vREX (Krueger et al., 2021) improves consistency by minimizing variance of losses across training domains.

(7) GroupDRO (Sagawa et al., 2019) aims to maintain good performance in the worst-case scenario by minimizing the maximum risk over a set of possible distributions, which uses domains as groups.

(8) ERM++ (Teterwak et al., 2025) optimizes original ERM on initialization and tuning to reduce overfitting.

For test-time training methods, we compare with

(1) TTT++ (Liu et al., 2021) manipulated the model in both the training and testing phases with Online Feature Alignment and Online Dynamic Queue.

(2) MT3 (Bartler et al., 2022) manipulated the model in both the training and testing phases by optimizing a meta-learning objective.

For missing robust methods, we compare with:

(1) DAMS-adjust (Zhou et al., 2023a) proposes domain adaptation under missingness shift by estimating relative missingness rates to recover the optimal target predictor.

(2) StableMiss (Zhu et al., 2023) approximates the invariant optimal predictors under different masks via double parameterization and enforces decorrelation between features and masks, thereby achieving robust prediction under agnostic mask distribution shift

For tabular large language models, we compare with

(1) TabLLM (Hegselmann et al., 2023) transforms structured tabular data into textual sequences and leverages pretrained language models for classification, enabling efficient and accurate tabular classification in low-resource (few-shot) scenarios.

(2) TableGPT2 (Su et al., 2024) is a unified fine-tuning framework that enables models to understand and manipulate tabular data using external functional commands.

## C.2  IMPLEMENTATIONS DETAILS

In this subsection, we provide the details of the backbone model and the configuration of the training and testing phases to enhance reproducibility. All experiments are conducted on a Linux server with NVIDIA GeForce RTX 4090 GPUs.

**Backbone Models** We use two deep tabular models for test-time training methods: MLP, TabM (Gorishniy et al., 2025) as the backbone models.

For our *KGCS4Tab* method, we use three tree-based tabular models, GBDT, CatBoost (Prokhorenkova et al., 2018), and LightGBM (Ke et al., 2017), and two deep tabular models, MLP and TabM (Gorishniy et al., 2025).

**Training Phase** For domain generalization methods, we use the hyperparameters provided in the TableShift benchmark (Gardner et al., 2023) if they have them or their hyperparameters in the paper.

For test-time training methods, we train each backbone model with a batch size of 512 (MLP), 128 (TabM) for several epochs, depending on the model's convergence as evaluated on the validation set. The AdamW optimizer is used with a learning rate of 0.01 and a weight decay of 0.01 for MLP. For TabM, the optimizer is constructed with the provided optimizer construction function, with a learning rate of 0.02 and a weight decay of 0.003.

For tabular large language models, we test TabLLM (Hegselmann et al., 2023) following the fine-tuning framework introduced in their paper. We test TableGPT2 (Su et al., 2024) without fine-tuning because the model has the zero-shot prediction ability according to their paper.

For *KGCS4Tab* methods, we divide label shift into 5% intervals with $n = 8$ levels, train a separate model for each, and use $\lambda = 0.8$ for probabilistic selection. The rule proposer is DeepSeek-V3 (Liu et al., 2024). We consider a rule valid if it achieves better performance than zero imputation on the in-distribution validation set. We train each model with a batch size of 512 (MLP), 128 (TabM) for several epochs, depending on the model's convergence as evaluated on the id-validation set. The AdamW optimizer is used with a learning rate of 0.01 and a weight decay of 0.01 for MLP. For TabM, the optimizer is constructed with the provided optimizer construction function, with a learning rate of 0.02 and a weight decay of 0.003. Other backbones follow the default hyperparameter of the TableShift benchmark. Baselines follow TABLESHIFT (Gardner et al., 2023) defaults. All experiments are run on Linux with NVIDIA 4090 GPUs.

**Testing Phase** For test-time training, we set a batch size of 64 for testing. For the *KGCS4Tab* method, we set the probability $\lambda = 0.8$. For all missing column, we use **Zero Imputation** by default.

## C.3 DATASET DETAILS

**ANES.** Understanding electoral participation is a critical task for policymakers, political strategists, and advocates of democratic engagement. Accurately predicting which individuals are likely to vote in an election is essential for effective polling and campaigning, particularly in U.S. politics. The American National Election Studies (ANES) dataset is designed to predict whether an individual will vote in a U.S. presidential election. It contains 60,377 samples and includes 365 features covering voting behavior, electoral history, public opinion, and political attitudes.

**Hypertension.** Hypertension—defined as systolic blood pressure of 130 mm Hg or higher, or diastolic pressure of 80 mm Hg or higher—affects nearly half of all Americans. If left untreated, it is one of the most significant risk factors for heart attack and other cardiovascular diseases. Therefore, accurate and efficient prediction of blood pressure levels is of great clinical importance. The Hypertension dataset aims to improve the effectiveness of blood pressure measurement and enhance prediction accuracy. It consists of 846,781 samples with 100 features related to various hypertension risk factors.

**ACSincome.** Income is a widely used indicator of social and economic stability. It serves as a key criterion in numerous social support programs. For example, in the United States, income levels are used to assess poverty status and determine eligibility for services such as food stamps and Medicaid. Income prediction not only has substantial commercial value but also holds historical importance in the machine learning community, with roots tracing back to the well-known "adult income" census dataset.

**ACSpubcov.** Public health insurance plays a vital role in ensuring affordable and accessible healthcare for individuals. A high level of insurance coverage is essential for personal well-being and public health. Therefore, increasing public health insurance participation is a key policy objective. The ACSpubcov dataset focuses on predicting whether an individual is covered by public health insurance, using 135 features. It contains a total of 5,916,565 samples.

## C.4 EXTENDED EXPERIMENTS ON OBSERVATION 1

The way of injecting distribution shift and feature missingness is: the distribution shift originates from real-world datasets. In the TableShift benchmark, each dataset is partitioned into a validation set without distribution shift and a test set with distribution shift and feature missingness is introduced based on the feature importance estimated by a GBDT model. When injecting missingness, we remove the feature with the highest importance score.

Observation 1 can be observed across most datasets, though the magnitude varies. The results in the figure present a more pronounced example. For instance, in the ANES dataset, the performance drop caused by distribution shift is 4.05, the drop caused by feature missingness is 0.04, and the coupled change results in a drop of 4.12. Therefore, this phenomenon is not an isolated case.

To validate that Observation 1 is not an isolated case, we also conduct experiments by introducing missingness to each of the top ten most important features. The averaged performance drops are 1.9, 0.7, and 3.2, respectively. On the Hypertension dataset, the corresponding averages are 6.6, 0.7, and 7.5. These results demonstrate that we observe the same phenomenon across multiple datasets and multiple features, which verifies that the finding is robust rather than anecdotal.

## C.5 DETAILED EXPERIMENT RESULTS IN RQ3

Table 10: Performance of *KGCS4Tab* approach and comparison methods on 4 datasets with 5 tabular backbones under 2 and 3 columns missing, the best performance is in **bold**, selected in Table 9 is underlined

| Methods | | ANES (2 columns) | | Hypertension (2 columns) | | ANES (3 columns) | | Hypertension (3 columns) | |
|---|---|---|---|---|---|---|---|---|---|
| | | ACC | F1 | ACC | F1 | ACC | F1 | ACC | F1 |
| **Fundamental Tabular Model** | | | | | | | | | |
| GBDT | | 75.36± 0.00 | 82.03± 0.00 | 45.64± 0.00 | 18.16± 0.00 | 72.10± 0.00 | 80.57± 0.00 | 44.84± 0.00 | 14.80± 0.00 |
| LightGBM | | 78.36±0.00 | 83.66±0.00 | 46.94±0.00 | 23.25±0.00 | 76.82±0.00 | 82.96±0.00 | 45.19±0.00 | 16.03±0.00 |
| CatBoost | | 79.39± 0.00 | 84.31± 0.00 | 48.86± 0.00 | 31.40± 0.00 | 77.63± 0.00 | 83.48± 0.00 | 47.34± 0.00 | 25.82± 0.00 |
| MLP | | 78.53 ± 0.43 | 83.81± 0.22 | 48.32 ± 1.28 | 29.84 ± 4.92 | 76.18 ± 0.58 | 82.66 ± 0.24 | 47.98 ± 0.60 | 28.81 ± 2.45 |
| TabM | | 79.75 ± 0.30 | 84.46 ± 0.12 | 48.51 ± 0.97 | 29.98 ± 3.36 | 77.79 ± 0.21 | 83.54 ± 0.10 | 45.59 ± 0.46 | 18.30 ± 1.93 |
| **Shift Robust Methods** | | | | | | | | | |
| ERM | | 77.45±0.28 | 83.35±0.11 | 45.06±4.93 | 14.60±20.64 | 77.39±1.11 | 83.12±0.43 | 47.19±7.95 | 24.59±34.77 |
| IRM | | 78.04±1.12 | 83.53±0.48 | 47.19±7.94 | 20.67±29.21 | 76.20±1.11 | 82.69±0.49 | 42.38±1.15 | 3.98±5.62 |
| vREX | | 76.04±1.86 | 82.67±0.91 | 42.21±0.90 | 3.13±4.42 | 77.51±0.35 | 83.21±0.11 | 41.57±0.00 | 0.00±0.00 |
| IB_IRM | | 78.23±1.69 | 83.46±0.62 | 42.53±1.35 | 4.65±6.54 | 76.41±1.51 | 82.77±0.63 | 44.02±2.14 | 11.68±10.00 |
| IB_ERM | | 78.12±1.06 | 83.61±0.45 | 42.60±1.37 | 5.02±6.68 | 76.11±0.75 | 82.64±0.33 | 41.64±0.05 | 0.33±0.26 |
| EQRM | | 76.57±2.44 | 82.71±1.18 | 53.09±8.15 | 46.06±32.79 | 75.51±0.41 | 82.30±0.18 | 52.81±7.95 | 49.17±34.77 |
| GroupDRO | | 78.17±0.91 | 83.56±0.29 | 41.57±0.00 | 0.00±0.00 | 76.96±0.81 | 83.01±0.34 | 43.40±2.46 | 8.66±11.63 |
| ERM++ | | 76.69±1.84 | 82.73±0.79 | 41.57±0.00 | 0.01±0.02 | 74.18±1.86 | 81.67±0.82 | 41.57±0.00 | 0.00±0.00 |
| MLP | TTT++ | 75.18 ± 1.36 | 82.24 ± 0.70 | 56.14 ± 1.79 | 58.47 ± 6.67 | 75.45 ± 0.35 | 82.34 ± 0.20 | 54.33 ± 0.44 | 55.10 ± 0.09 |
| | MT3 | 79.93 ± 0.30 | 84.48 ± 0.02 | 42.12 ± 0.39 | 2.70 ± 1.91 | 78.54 ± 0.41 | 83.84 ± 0.09 | 41.88 ± 0.22 | 1.51 ± 1.07 |
| TabM | TTT++ | 77.81 ± 3.63 | 83.50 ± 1.72 | 43.23 ± 2.35 | 7.70 ± 10.89 | 76.27 ± 3.52 | 82.71 ± 1.65 | 43.24 ± 2.37 | 8.21 ± 11.61 |
| | MT3 | 78.02 ± 0.28 | 83.77 ± 0.14 | 41.76 ± 0.10 | 0.97 ± 0.48 | 75.07 ± 0.29 | 82.28 ± 0.15 | 41.69 ± 0.07 | 0.62 ± 0.36 |
| **Missing Robust Methods** | | | | | | | | | |
| DAMS | | 60.76±0.00 | 40.76±0.00 | 58.42±0.00 | 36.88±0.00 | 63.03±0.00 | 46.34±0.00 | 58.42±0.00 | 36.88±0.00 |
| StableMiss | | 59.44± 0.00 | 37.28± 0.00 | 41.57± 0.00 | 29.36± 0.00 | 59.44± 0.00 | 37.28± 0.00 | 41.57± 0.00 | 29.36± 0.00 |
| **Tabular Large Language Models** | | | | | | | | | |
| TabLLM | | 45.91± 0.00 | 38.82± 0.00 | 52.45±0.00 | 58.53± 0.00 | 46.23± 0.00 | 40.24± 0.00 | 51.61± 0.00 | 61.81±0.00 |
| TableGPT2 | | 48.49± 0.00 | 49.34± 0.00 | 47.92± 0.00 | 48.00± 0.00 | 47.88± 0.00 | 46.47± 0.00 | 46.91± 0.00 | 43.20± 0.00 |
| **Ours** | | | | | | | | | |
| GBDT+Ours | | 79.88 ± 0.17 | 84.50 ± 0.02 | 60.82 ± 0.62 | 65.80 ± 1.73 | 79.43 ± 0.10 | 84.23 ± 0.08 | 60.23 ± 0.56 | 64.43 ± 1.53 |
| LightGBM+Ours | | 80.33 ± 0.08 | 84.60 ± 0.06 | **61.10 ± 0.49** | **66.23 ± 1.39** | 80.22 ± 0.10 | 84.55 ± 0.10 | **60.60 ± 0.49** | **64.94 ± 1.39** |
| CatBoost+Ours | | **80.41 ± 0.14** | **84.64 ± 0.06** | 60.71 ± 0.50 | 65.67 ± 1.39 | 80.16 ± 0.05 | 84.48 ± 0.09 | 60.31 ± 0.47 | 64.71 ± 1.37 |
| MLP+Ours | | 79.35 ± 0.31 | 84.06 ± 0.14 | 52.32 ± 3.40 | 42.25 ± 10.81 | 79.23 ± 0.28 | 84.08 ± 0.07 | 53.05 ± 1.08 | 45.51 ± 3.17 |
| TabM+Ours | | 80.26 ± 0.09 | 84.59 ± 0.04 | 52.82 ± 7.93 | 49.24 ± 34.65 | 80.10 ± 0.19 | **84.60 ± 0.08** | 47.20 ± 7.93 | 24.70 ± 34.67 |

## C.6 ROBUSTNESS OF HYPERPARAMETERS

To validate whether our proposed *KGCS4Tab* approach is robust to the choices of hyperparameters, we conduct hyperparameter robustness experiments. We select the hyperparameter in the probabilistic selection strategy $\lambda$ to the value in $\{0.95, 0.9, 0.85, 0.8, 0.75\}$. As shown in Fig.10 and 11, the performance of *KGCS4Tab* keep stable when $\lambda$ changes. The results demonstrate that *KGCS4Tab* is robust to slight changes in all hyperparameters.

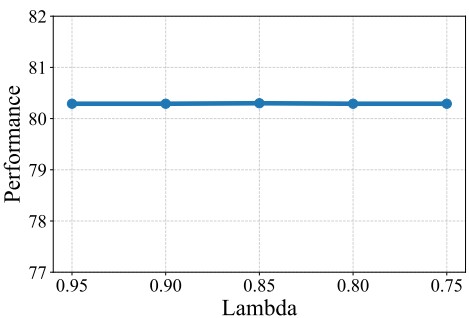

Figure 10: Performance of ANES dataset with different $\lambda$

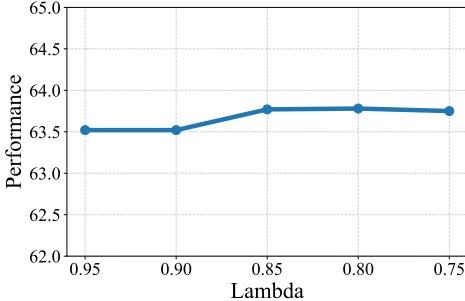

Figure 11: Performance of Hypertension dataset with different $\lambda$

## C.7 COMPARISON BETWEEN OUR METHOD AND DIRECT ENSEMBLE METHODS

We conducted experiments with direct ensemble models in KGFA and compared them with our results. In most cases, our selection strategy outperforms simple ensembling, highlighting the effectiveness of our approach.

Table 11: Comparison of our method with ensemble methods

| Model | ANES | Hypertension | Acsincome | Acspubcov |
|---|---|---|---|---|
| CatBoost | 79.86 | 58.32 | 76.42 | 67.14 |
| CatBoost+Ensembling | 80.39 | 60.51 | 81.07 | 68.95 |
| CatBoost+Ours | **80.41** | **63.46** | **81.21** | **74.55** |
| LightGBM | 79.87 | 56.48 | 76.74 | 64.68 |
| LightGBM+Ensembling | 80.24 | 60.75 | 80.50 | 66.15 |
| LightGBM+Ours | **80.38** | **63.75** | **80.86** | **73.59** |

# D DISCUSSION ABOUT PROPOSED RULES

## D.1 THE QUALITY OF THE RULES PROPOSED BY KGFA

To assess the quality of the rules proposed by $\mathcal{P}$, we evaluate it by computing the performance improvement in the validation set compared to zero imputation. In our experiments, we accept a rule if its performance on the in-distribution validation set surpasses zero imputation, indicating that it contributes more useful information. We further analyze how rule quality relates to overall *KGCS4Tab* performance (Performance improvement in Fig 12), where rule quality is measured by the improvement over zero imputation in the in-distribution validation set (Rule quality in Fig 12). As shown in Fig. 12, as the quality increases, the performance improvement also increases. Although the degree of improvement is influenced by multiple factors, these results indicate that our measurement of rule quality is reasonable, and our method can potentially mitigate the risk of erroneous rules in practice. Our experimental results also show that, in our experiments, the rules proposed by the LLM are indeed effective, further demonstrating the effectiveness of our approach.

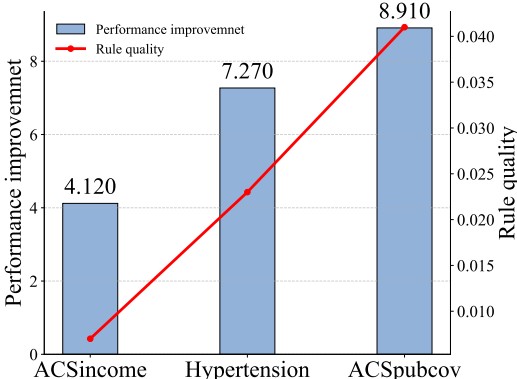

Figure 12: The relationship between the quality of rules and performance improvement

## D.2 Rule Correlations and Performance Recovery Ratios

We employed a decision tree model to perform a recovery task on the missing columns in the training set, thereby obtaining the importance of each feature during the recovery process to verify the correlation between our proposed rules and the existing features. We then compared the importance of the features used in our recovery rules with the total importance of all features to calculate the percentage of importance they account for. The experimental results are as follows:

Table 12: Rule Correlations on different datasets

| Model | ANES | Hypertension | acsincome | acspubcov |
|-------|------|--------------|-----------|-----------|
| Ours | 98.25% | 81.49% | 63.18% | 64.83% |

As can be seen, the features used in our proposed rules account for a substantial portion of the overall importance, indicating that our strategy can effectively capture the correlations. It is worth noting that in datasets such as ANES, the column names are mostly coded (e.g., "VCFxxx") and contain limited information, whereas in datasets like Hypertension, column names such as "DRNK_PER_WEEK" carry more semantic meaning. Therefore, our evaluation encompasses both cases: columns with inherent correlation in their names and those without.

Meanwhile, regarding the question on the upper bound for performance gains, we computed the performance using the fully correct test distribution (rather than using DAMS to estimate the ground truth) and then calculated the proportion relative to our method's performance. This allows us to represent the upper bound of performance recovery achievable by our approach, as shown in the table below:

Table 13: Recovery Ratios across datasets

| Model | ANES | Hypertension | acsincome | acspubcov |
|-------|------|--------------|-----------|-----------|
| LightGBM+gt | 80.40 | 63.89 | 81.43 | 75.19 |
| LightGBM+Ours | 80.38 | 63.75 | 80.86 | 73.59 |
| ratios | 99.97% | 99.78% | 99.30% | 97.87% |

As shown in the table, our method achieves a high proportion of performance recovery relative to the ground truth. By adopting our selection strategy, we obtain a balance between efficiency and accuracy, which demonstrates the effectiveness of our approach.

# E    TIME AND MEMORY COST

Our methods need to train multiple models. In this section, we will discuss the usage of memory and time.

Regarding storage efficiency, tabular prediction models are typically very compact. For instance, on the ACSpubcov dataset, our method's total model storage is only 9584KB, which represents merely 0.5% of the dataset size (1,831,301KB). This demonstrates that the additional storage overhead is negligible and does not impose any significant burden.

Regarding time efficiency, our approach allows for parallel training of multiple models, because there are no dependencies between the training processes of different models. This parallel implementation effectively ensures that the total training time remains comparable to that of training a single model, thus minimizing computational overhead.

It should be noted that when the missing columns are unknown, we can prompt the construction of imputation rules for each column and train using our method. This is feasible in practical deployment, as our approach offers advantages in both storage and computation time. Since detecting entirely missing columns is relatively straightforward—requiring only counting the number of NaNs in each column—our method can still be effective even when the missing columns are unknown. Also, as proposing a rule only needs 2 calls to the LLM api, it will not cause a high compute burden.

## F   Failure Mode of LLMs

As mentioned in the Limitation section, our KGFA proposes a recovery rule based on the semantic background of the dataset. Therefore, it may be invalid when all the semantic information cannot be obtained. In this situation, we can prompt the LLMs to use the statistical information to propose imputation rules.

As for the failure of the proposed rules, we give a case study. This proposed rule is not accepted by our methods because it cannot surpass zero imputation in the validation set.

---

**A fail case**

To recover the missing values of the 'AGEG5YR' column, we can create a rule based on employment-related columns that are strongly correlated with age. The 'AGEG5YR' column typically represents age groups in 5-year intervals. Since employment status often varies by age—young adults may be entering the workforce, middle-aged adults are usually fully employed, and older adults may be partially employed or retired—employment information can serve as a proxy for age groups.

### Proposed Rule:

We can use the following columns to approximate 'AGEG5YR':

1. **EMPLOY1_10**: Represents employment counts in a younger age subgroup. 2. **EMPLOY1_40**: Represents employment counts in a middle-aged subgroup. 3. **EMPLOY1_60**: Represents employment counts in an older age subgroup.

By summing these columns, we can recover 'AGEG5YR' as a combined indicator of the overall age distribution in the dataset.

### Steps to Recover 'AGEG5YR':

1. **Use Employment Subgroups as Age Indicators**:

* 'EMPLOY1_10' → younger adults * 'EMPLOY1_40' → middle-aged adults * 'EMPLOY1_60' → older adults

2. **Sum the Subgroups**:

```python
df['AGEG5YR'] = df['EMPLOY1_10'] + df['EMPLOY1_40'] + df['EMPLOY1_60']
```

This generates a combined feature reflecting the overall age group distribution.

3. **Rationale**:

* Employment status is closely linked to age. * Summing these subgroups captures contributions from all relevant age ranges. * This method is simple, interpretable, and avoids complex imputation models. * It allows indirect recovery of missing 'AGEG5YR' values using already available information in a logically consistent way.

---

This case illustrates a failure of LLMs to fully comprehend the column description. Specifically, the model treats EMPLOY1 as a single homogeneous group and simply sums its values, whereas in reality, EMPLOY1 is closely related to age and should be processed separately for different age segments. Such misunderstandings can be empirically detected through our validation performance, which demonstrates that our method reliably identifies and corrects these errors, highlighting its robustness and effectiveness in handling nuanced feature relationships.

# G  CASE STUDY

## G.1  PROMPT TEMPLATE OF RULE PROPOSER

We first introduce the prompts used in KGFA to propose rules by the guidance of the background.

**Rule Proposer Prompt**

You are a helpful assistant

### Task ###

There is a tabular dataset related with {Background $\mathcal{B}$}. Your objective is to predict {Goal $\mathcal{G}$}. You have access to the following attributes: {Column semantic descriptions $\mathcal{M}$}

To enhance prediction performance in real world applications, you need to propose rule to recover the value of column {$c_m$} using other columns to overcome the situation that when this columns value is missing when testing. The rule you designed should be reasonable to its original meaning, for example, for the original value, if it is the bigger the better, then the recovered value should have the same properties. You can change the value to catgorial with the value has same trend to original value. The rule you designed will be directly used to recover the whole column. You need to covert all the value to a number which can be direct used by machine learning model.

### Answer ###

**Proposed Rule to Code Prompt**

You are a helpful assistant

### Task ###

We have a rule: {rule $r$}, covert it to python code, the input parameter is only the test pd.Dataframe, we want the rule can be directly used with the function. The function name should be recover, it need to be used as test_df = recover(test_df). You need to just give me the code which can be directly use exec to run, no other imformation.

### Answer ###

## G.2 EXAMPLE OF PROPOSED RULES

In this section, we present one original response following the rule proposer prompt, and four code responses following the code prompt.

---

**Original response in Hypertension dataset**

To recover the missing values of the 'AGEG5YR' column, we need to create a rule based on other available columns that can reasonably approximate age groups. The 'AGEG5YR' column typically represents age groups in 5-year intervals (e.g., 18-24, 25-29, etc.).

### Proposed Rule: We can use the following columns to approximate 'AGEG5YR': 1. **IYEAR (Year of Interview)**: This can be used to infer the approximate age if combined with other health-related behaviors.

2. **DRNK_PER_WEEK (Drinks per week)**: Drinking habits can vary by age group.

3. **SMOKE100 (Smoked at least 100 cigarettes in life)**: Smoking behavior is often age-dependent.

4. **TOTINDA (Physical activity)**: Physical activity levels can vary by age.

5. **EMPLOY1 (Employment status)**: Employment status can hint at age (e.g., retired vs. working).

### Steps to Recover 'AGEG5YR': 1. **Categorize Numerical Columns**:
- 'DRNK_PER_WEEK':
- Low (0-7 drinks/week): Younger or older adults.
- Medium (8-14 drinks/week): Middle-aged adults.
- High (15+ drinks/week): Younger or middle-aged adults.
- 'IYEAR':
- Recent years (2020-2022) might imply younger participants.
- Older years (2015-2019) might imply older participants.
2. **Use Categorical Columns**:
- 'SMOKE100_10' (Yes): More common in older adults.
- 'SMOKE100_20' (No): More common in younger adults.
- 'TOTINDA_10' (Physically active): More common in younger adults.
- 'TOTINDA_20' (Not physically active): More common in older adults.
- 'EMPLOY1_10' (Employed): Likely middle-aged.
- 'EMPLOY1_70' (Retired): Likely older adults.
3. **Assign Age Groups**:
- If 'EMPLOY1_70' is 1 (Retired): Assign to the highest age group (e.g., 65+).
- If 'SMOKE100_10' is 1 and 'TOTINDA_20' is 1: Assign to middle-aged groups (e.g., 45-64).
- If 'SMOKE100_20' is 1 and 'TOTINDA_10' is 1: Assign to younger groups (e.g., 18-44).
- Adjust based on 'DRNK_PER_WEEK' and 'IYEAR' for finer granularity.
4. **Convert to Numerical Value**:
- Map age groups to numerical values (e.g., 1 for 18-24, 2 for 25-29, ..., 13 for 65+).

---

**Proposed Rule for ANES dataset**

```python
import pandas as pd

def recover(test_df):
    def recover_VCF0717_00(row):
        if (row.get('VCF0803_20', 0) == 1 or row.get('
            VCF0301_30', 0) == 1):
            if (row.get('VCF0803_10', 0) == 0 and row.get
                ('VCF0803_30', 0) == 0 and
                row.get('VCF0301_00', 0) == 0 and row.get
                    ('VCF0301_10', 0) == 0 and
                row.get('VCF0301_40', 0) == 0 and row.get
                    ('VCF0301_50', 0) == 0):
                return 1
        return 0

    if 'VCF0717_00' in test_df.columns:
        test_df['VCF0717_00'] = test_df.apply(
            recover_VCF0717_00, axis=1)
    return test_df
```

Listing 1: Proposed Rule for ANES dataset

**Proposed Rule for Hypertension dataset**

```python
import numpy as np
import pandas as pd

def recover(test_df):
    def recover_age(row):
        if 'EMPLOY1_80' in row and row['EMPLOY1_80'] ==
            1:
            return 1.55
        baseline = 0.0 if 'EMPLOY1_10' in row and row['
            EMPLOY1_10'] == 1 else -0.5
        diabetes = row['DIABETES'] if 'DIABETES' in row
            else 0
        totinda = 1 - row['TOTINDA_10'] if 'TOTINDA_10'
            in row else 0
        drinks = row['DRNK_PER_WEEK'] if 'DRNK_PER_WEEK'
            in row else 0
        poverty = row['POVERTY'] if 'POVERTY' in row else
             0
        age = baseline + diabetes*0.5 + totinda*0.3 -
            drinks*0.01 + poverty*0.2
        return np.clip(age, -1.52, 1.55)

    if 'AGEG5YR' in test_df.columns:
        test_df['AGEG5YR'] = test_df.apply(recover_age,
            axis=1)
    else:
        test_df.insert(0, 'AGEG5YR', test_df.apply(
            recover_age, axis=1))
    return test_df
```

Listing 2: Proposed Rule for Hypertension dataset

**Proposed Rule for ACSincome dataset**

```python
import pandas as pd

def recover(test_df):
    def recover_wkhp(row):
        if 'WRK_00' in row and row['WRK_00'] == 1:
            return 0
        elif 'WKW' in row and row['WKW'] < 0.5:
            return 0
        elif ('OCCP_PRT' in row and row['OCCP_PRT'] == 1)
                or ('OCCP_EAT' in row and row['OCCP_EAT'] ==
                1):
            return 1
        elif 'WKW' in row and row['WKW'] >= 1.5 and (('
                OCCP_MGR' in row and row['OCCP_MGR'] == 1) or
                ('OCCP_ENG' in row and row['OCCP_ENG'] == 1)
                ):
            return 3
        else:
            return 2

    if 'WKHP' not in test_df.columns:
        test_df['WKHP'] = test_df.apply(recover_wkhp,
                axis=1)
    else:
        test_df['WKHP'] = test_df.apply(lambda row:
                recover_wkhp(row) if pd.isna(row['WKHP'])
                else row['WKHP'], axis=1)
    return test_df
```

Listing 3: Proposed Rule for ACSincome dataset

**Proposed Rule for ACSpubcov dataset**

```python
import pandas as pd

def recover(test_df):
    def recover_AGEP(row):
        # Step 1: Assign base AGEP based on SCHL
        if any(row[f"SCHL_{i:02d}"] == 1 for i in range
            (1, 10)):
            agep = -1.0
        elif any(row[f"SCHL_{i:02d}"] == 1 for i in range
            (10, 17)):
            agep = 0.0
        else:
            agep = 1.0

        # Step 2: Adjust for MAR
        if row["MAR_01"] == 1:
            agep += 0.2
        elif row["MAR_02"] == 1:
            agep -= 0.2

        # Step 3: Adjust for PINCP
        agep += row["PINCP"] * 0.3

        # Step 4: Clamp to original range
        agep = max(-1.314, min(1.879, agep))
        return agep

    test_df['AGEP'] = test_df.apply(recover_AGEP, axis=1)
    return test_df
```

Listing 4: Proposed Rule for ACSpubcov dataset

## H  LLM USAGE

In this work, we used large language models (LLMs) only in a limited, supportive capacity. All key research ideas, theoretical analyses, experimental designs, and the writing of the main text were carried out independently by the authors. LLMs were not used to generate or edit any scientific content in the manuscript, nor did they contribute to the formulation of research hypotheses or the interpretation of results. The authors take full responsibility for the accuracy, originality, and completeness of all content in the paper.

The use of LLM is only about aiding or polishing writing in this paper.

It mainly has two purposes in usage, with the prompt below.

First one is polishing writing:

**Prompt to Polish**

Help me to polish the sentence below: {sentence}

The second one is to write concisely to satisfy the space limit in ICLR:

**Prompt for Concise Writing**

Help me write the sentence below concisely: {sentence}

All the polished representations obtained from LLMs are double-checked by the authors of this paper.

