# OpenReview forum: "Handling Tabular Data under Coupled Shifts of Feature Missingness and Distributional Change"
_ICLR.cc/2026/Conference — Submitted to ICLR 2026_

### Official Review · Reviewer_nxRD · 2025-10-24

**Soundness:** 4
**Presentation:** 4
**Contribution:** 2
**Rating:** 6
**Confidence:** 4

**Summary:**

The paper addresses the problem of robust tabular learning under the combined challenges of missing features and distribution shifts in test data. The authors propose a two-module solution: a knowledge-guided feature aligner (KGFA) that imputes the missing feature via proposing imputation rules, and a distribution-aware model-selection (DAMS) module that selects the most appropriate model based on the detected distribution shift. The proposed method is evaluated on several benchmark datasets, demonstrating its effectiveness compared to existing baselines.

**Strengths:**

1. The paper points a realistic problem in the tabular learning area, where the test data is having both missing features and distribution shifts. This would cause tabular model performance degradation and none of the existing robustness methods can handle well.
2. The paper proposes a systematical solution to address the problem, which includes a missing-feature imputation module and a distribution-aware model-selection module. The two modules are well designed and supported with theoretical fundations.
3. The experimental results on various baselines demonstrate the effectiveness of the proposed method.

**Weaknesses:**

1. Although the impact of the proposed problem is proven by experiments, the novelty of the problem itself is limited. In other words, suppose the missing feature is problem A, and the distribution shift is problem B. Then if there is a new problem C called "new features appearing in test data" coming up, then we can create a list of problem combinations of AB, AC, BC, and ABC. Therefore, the proposed problem is just one of many possible combinations of existing problems.
2. The proposed method tries to train multiple models for different possible feature missing situations, which may not be scalable when the number of possible missing columns is large

**Questions:**

1. In Fig.9, the paper compares the proposed method with LLM based baselines, which is an odd choice since CatBoost and XGBoost are generally performing better than LLMs on tabular data. What is the reason for choosing LLM baselines here? Are there any results of comparing with CatBoost/XGBoost on the same setting?
2. For the ablation study, how is without DAMS implemented? Is it using a single model trained on complete data and testing on missing+shifted data? How is the performance of ensembling multiple models without DAMS (e.g., averaging the outputs of all models) compared to using DAMS?

---

> ### Author Response · Authors · 2025-11-15
>
> We sincerely appreciate your valuable feedback on our work. We address each of your concerns in detail below.
> ***
> [W1] Our work focuses on the coupled effects caused by distribution shift and feature missingness. As analyzed in the paper, addressing distribution shift or feature missingness individually is insufficient to tackle this problem. Therefore, this is not a simple A+B scenario: the combination of multiple challenges introduces new difficulties that do not arise in a mere A+B setting.
> ***
> [W2] Since our method incurs relatively low storage and computational overhead (Appendix E), it remains effective even when a large number of columns may be missing.
> ***
> [Q1] We chose to compare with LLM-based methods because they leverage the background knowledge of LLMs, which may provide better robustness to coupled shifts. The experimental results, where TabLLM achieves relatively strong performance on some datasets, also support this point. Under the same settings, CatBoost runs faster than our method, as our approach introduces additional distance computations.
> ***
> [Q2] The implementation of “without DAMS” is as follows: the model is trained on the augmented training set and then tested on data with missing values and distribution shifts. In this variant, there is no model selection stage; the model that aligns with the training distribution is automatically used for prediction. Compared to the full method, it lacks the process of selecting models based on distribution-aware computations.
>
> We conducted experiments with direct ensemble methods and compared them with our results. In most cases, our selection strategy outperforms simple ensembling, highlighting the effectiveness of our approach.
>
> | Model               | ANES    | Hypertension | Acsincome | Acspubcov |
> |---------------------|---------|--------------|-----------|-----------|
> | CatBoost            | 79.86   | 58.32        | 76.42     | 67.14     |
> | CatBoost+Ensembling | 80.39   | 60.51        | 81.07     | 68.95     |
> | CatBoost+Ours       | **80.41**   | **63.46**        | **81.21**     | **74.55**     |
> | LightGBM            | 79.87   | 56.48        | 76.74     | 64.68     |
> | LightGBM+Ensembling | 80.24   | 60.75        | 80.50     | 66.15     |
> | LightGBM+Ours       | **80.38**   | **63.75**        | **80.86**     | **73.59**     |

---

> > ### Comment · Reviewer_nxRD · 2025-11-24
> >
> > I appreciate the authors for the feedback. My concerns on Q1, Q2 are addressed. I have an additional question on my W2, that is the cost of calling LLMs. It is better to analyze how frequently LLMs are called to finish the rule proposal, which is also crucial for scalability.

---

> > > ### Author Response · Authors · 2025-11-24
> > >
> > > Thank you for your response. I’m glad we were able to address your concerns, and your feedback is very helpful for improving our work.
> > > ***
> > > [W2] Regarding the issue you mentioned about the frequency of invoking the LLM, our method requires only two LLM calls to generate a recovery rule: one to analyze the dependencies between columns and propose a recovery rule, and another to translate the rule into executable code. In our experimental environment, the total time for two API calls to DeepSeek-V3 is under 30 seconds.
> > > Therefore, even when many columns are missing, the computational overhead remains low, and the method is highly scalable.

---

### Official Review · Reviewer_6gUQ · 2025-10-27

**Soundness:** 2
**Presentation:** 2
**Contribution:** 2
**Rating:** 4
**Confidence:** 5

**Summary:**

The paper introduces the RTCS problem — Robust Tabular prediction under Coupled Shifts — where both: Feature missingness (entire feature columns missing at test time), and Distribution shifts (differences between training and test distributions) occur simultaneously. To mitigate it, it devises a KGFA(knowledge-guided feature aligner) and DAMS (model selector). Overall, the method shows reasonable gains among most datasets,most pronounced upon Hyperextension, for almost all tabular backbone models.

**Strengths:**

1. Use of external knowledge
* using "interpretable" external imputation rules seem quite novel in tabular data. Furthermore, validating the rules with a held-out distribution (almost) guarantees its effect upon mild test distributions
2. Theoretical guarantee
* theoretical guarantee upon the DAMS's efficient compute of distance.

**Weaknesses:**

1. Lack of comparisons with Imputation methods utilizing LLMs
* similar to that of the proposed method, there has been spark of works which have performed imputations using LLMs. specifically, https://openreview.net/pdf?id=8F6bws5JBy. I wish the authors could provide (if not this paper) additional comparisons of performance between 1. imputation + model inference/training vs. 2. proposed method, to further solidify its core idea.
2. Lack of analysis of "upper bound"
* both DAMS and RTCS seem to have an "upper bound", due to its 1. human interpretable nature and 2. the efficacy scheme. However, the authors did not conduct any analysis upon this. I wish the authors could create a synthetic dataset, with "predefined correlation rules", and check if the "rule proposer" can correctly grasp the correlation in both : 1) column names with inherent correlation in their naming (i.e. BMI & height) and 2) without it. 2) is necessary, since it is not always the case where the columns are that easy to comprehend & reason their correlation upon - for example, the stats of the gyroscope sensor where the column names are X, Y and Z.  Furthermore, for the DAMS, the authors should provide the upperbound for performance gains when model selections were done upon "ground-truth" distribution distance matrix, and up to how much percentage does DAMS recover, with its efficiency tradeoff.

If both these concerns are resolved, I'm willing to raise my score.

**Questions:**

please refer to the weakness.

---

> ### Author Response · Authors · 2025-11-15
>
> We sincerely appreciate your valuable feedback on our work. We address each of your concerns in detail below.
> ***
> [W1] Thank you for your suggestion. The LITO method you mentioned effectively addresses class imbalance in tabular prediction using LLMs and sampling techniques, providing valuable insights for researchers. We greatly appreciate this work and will introduce it in the related work section. Regarding the comparison with imputation + model inference/training that you mentioned, since your approach does not focus on feature missingness, we instead provide comparisons with traditional baselines.
>
> | Model          | ANES    | Hypertension | ACSincome | ACSpubco |
> |----------------|---------|--------------|-----------|----------|
> | LightGBM       | 79.87   | 56.48        | 76.74     | 64.68    |
> | MICE           | 79.85   | 59.73        | 79.36     | 61.61    |
> | LR             | 79.78   | 59.57        | 80.04     | 65.03    |
> | LightGBM+Ours  | **80.38**   | **63.75**        | **80.86**     | **73.59**    |
>
> Additionally, on TabFSBench, as a rough comparison, our method outperforms the ATLLM imputation approach on the eyemovements dataset (*the data is taken from ATLLM’s performance figures; the settings may differ slightly, so this is only an approximate comparison. ATLLM does not provide absolute performance on the jannis and concrete datasets, so comparison is only possible on this dataset). This set of experimental results demonstrates the effectiveness of our proposed method. We sincerely appreciate your insights.
>
> | Model           | eyemovements (multi-class)-ACC |
> |-----------------|--------------------------------|
> | LightGBM        | 64.21                          |
> |  ATLLM          |  around 65*                    |
> | LightGBM+Ours   | **68.00**                      |
>
> ***
> [W2] Regarding the comment on the predefined correlation rules, we designed the following experiment to verify the correlation between our proposed rules and the existing features. Specifically, we employed a decision tree model to perform a recovery task on the missing columns in the training set, thereby obtaining the importance of each feature during the recovery process. We then compared the importance of the features used in our recovery rules with the total importance of all features to calculate the percentage of importance they account for. The experimental results are as follows:
>
> | Model | ANES    | Hypertension | acsincome | acspubcov |
> |-------|---------|--------------|-----------|-----------|
> | Ours  | 98.25%  | 81.49%       | 63.18%    | 64.83%    |
>
> As can be seen, the features used in our proposed rules account for a substantial portion of the overall importance, indicating that our strategy can effectively capture the correlations. It is worth noting that in datasets such as ANES, the column names are mostly coded (e.g., "VCFxxx") and contain limited information, whereas in datasets like Hypertension, column names such as "DRNK_PER_WEEK" carry more semantic meaning. Therefore, our evaluation encompasses both cases: columns with inherent correlation in their names and those without.
>
> Meanwhile, regarding the question on the upper bound for performance gains, we computed the performance using the fully correct test distribution (rather than using DAMS to estimate the ground truth) and then calculated the proportion relative to our method’s performance. This allows us to represent the upper bound of performance recovery achievable by our approach, as shown in the table below:
>
> | Model          | ANES    | Hypertension | acsincome | acspubcov |
> |----------------|---------|--------------|-----------|-----------|
> | LightGBM+gt    | 80.40   | 63.89        | 81.43     | 75.19     |
> | LightGBM+Ours  | 80.38   | 63.75        | 80.86     | 73.59     |
> |Recovery  ratio         | 99.97%  | 99.78%       | 99.30%    | 97.87%    |
>
>
> As shown in the table, our method achieves a high proportion of performance recovery relative to the ground truth. By adopting our selection strategy, we obtain a balance between efficiency and accuracy, which demonstrates the effectiveness of our approach.
>
>
> ***
> Ref:
>
> [1] LITO: Language-interfaced tabular oversampling via progressive imputation and self-authentication. In: ICLR 2024
>
> [2] TabFSBench: TabFSBench: Tabular Benchmark for Feature Shifts in Open Environments. In: ICML 2025
>
> [3] ATLLM: Fully Test-Time Adaptation for Feature Decrement in Tabular Data. In: IJCAI 2025

---

### Official Review · Reviewer_Q3ez · 2025-10-30

**Soundness:** 2
**Presentation:** 3
**Contribution:** 2
**Rating:** 4
**Confidence:** 5

**Summary:**

This paper proposes KGCS4Tab, a framework addressing robust tabular prediction under coupled feature missingness and distribution shifts. It integrates large language models for knowledge-guided imputation and a distribution-aware selector for adaptive model choice. The formulation of the coupled-shift problem is novel, and the method achieves consistent empirical gains across several benchmarks. Overall, it’s an ambitious and creative step toward robust tabular learning in open environments.

**Strengths:**

- This paper defines a new and practical problem setting (RTCS) where missingness and distribution shift coexist, highlighting their compounded effect.

- It also introduces a novel combination of LLM-guided imputation and probabilistic model selection, linking knowledge reasoning with robustness.

- KGCS4Tab provides solid empirical results showing consistent gains across datasets and backbones.

- This paper also includes theoretical grounding and ablation studies that support the design choices and demonstrate the complementarity of modules.

**Weaknesses:**

### **Problem & Motivation**
- Figure 2 presents an interesting observation that the degradation under coupled shifts exceeds the additive effect of each individual shift. However, it is unclear whether this property consistently holds across all datasets or was selectively demonstrated. More clarification is needed on how the distribution shifts and missingness were injected to verify that the finding is robust rather than anecdotal.

- The related-work section divides distribution-shift robustness into domain generalization and test-time adaptation, but omits domain adaptation, which is one of the core paradigms in this area. A short discussion connecting domain adaptation to the proposed problem setting would make the taxonomy more complete.

- The conceptual separation between feature missingness and distribution shift also needs stronger justification. Once missing features are imputed, the resulting data distribution effectively changes, making the two phenomena difficult to treat as orthogonal. A more principled explanation or theoretical framing for why they should be considered distinct would strengthen the motivation.

- The assumption that entire columns are missing seems restrictive. In most real-world cases, only partial feature missingness occurs. It would be valuable to know whether the method also performs well under partial missingness and how sensitive it is to the degree of missingness.

- Overall, while the paper identifies a practical and underexplored problem, a more fundamental and thorough analysis of the coupled effect—beyond the three empirical observations—would make the problem formulation more compelling.

---

### **Method**

- The method treats each column independently when estimating distribution distances or generating imputation rules, which is questionable because tabular features are often correlated. Ignoring these inter-feature dependencies could limit the robustness of the approach in realistic settings.

- LightGBM is used as the primary base learner, though CatBoost generally achieves stronger performance on categorical tabular data. Explaining this design choice would make the experimental setup easier to interpret.

- The framework depends heavily on a large language model, yet the type of model and its prompting strategy are not clearly specified. Since performance could vary significantly across LLMs, the paper should explicitly state which model was used and whether different LLMs lead to consistent behavior.

---

### **Experiments**

- All experiments appear to inject missingness artificially. It would strengthen the contribution to include a setting where missingness occurs naturally in real data, since the current setup mainly verifies synthetic conditions.

- The experiments focus solely on binary classification tasks. Evaluating the framework on multiclass or regression problems would demonstrate broader applicability.

- The relatively small gain on the ANES dataset remains unexplained. The text attributes it to lower feature discriminative power, but the claim is not quantitatively supported. Providing evidence, such as feature importance or variance analysis, would make the argument convincing.

- Because the method includes an imputation module, it would be appropriate to evaluate its imputation quality directly. Comparing against common imputation baselines such as mean, k-NN, MICE, or GAIN and reporting reconstruction error or predictive performance would clarify where the improvement originates.

- Finally, the experiments assume that the identity of the missing column (c_m) is known in advance. This assumption risks data leakage and may not hold in real deployments. The authors should clarify whether (c_m) is predefined or detected automatically, and discuss the implications if this information is unavailable.

**Questions:**

- In Table 1, the authors use IRM as the representative distribution-shift baseline, but it is not obvious why this particular method was chosen over other general baselines such as DRO, EQRM, or ERM++. It would be helpful to explain whether the result generalizes across these variants or is specific to IRM.

- The description of the Knowledge-Guided Feature Aligner is also too abstract. The definition of the imputation rule (r) lacks sufficient detail on its structure, input–output form, and examples. Showing only a single rule in Figure 5 does not adequately illustrate how such rules are generated or applied. Including a concrete prompt example or pseudocode would clarify this process.

- The argument that (r) cannot be directly applied to the test set due to a “semantic gap” is difficult to understand. The paper should elaborate on what this semantic gap entails and why it prevents the straightforward application of the learned rules.

---

> ### Author Response · Authors · 2025-11-15
> **Problem & Motivation Section**
>
> We sincerely appreciate your valuable feedback on our work. We address each of your concerns in detail below.
> ***
> [P&M1] This phenomenon can be observed across most datasets, though the magnitude varies. The results in the figure present a more pronounced example. For instance, in the ANES dataset, the performance drop caused by distribution shift is 4.05, the drop caused by feature missingness is 0.04, and the coupled change results in a drop of 4.12. Therefore, this phenomenon is not an isolated case.
>
> Regarding the injection of distribution shift and feature missingness:
>
> Distribution shift: The distribution shift originates from real-world datasets. In the TableShift benchmark, each dataset is partitioned into a validation set without distribution shift and a test set with distribution shift.
>
> Feature missingness: Feature missingness is introduced based on the feature importance estimated by a GBDT model. When injecting missingness, we remove the feature with the highest importance score.
>
> On the ACSIncome dataset, we conducted experiments by introducing missingness to each of the top ten most important features.  We show the performance drop under distribution shifts, feature missingness, and coupled shifts. The averaged performance drops are 1.9, 0.7, and 3.2, respectively. On the Hypertension dataset, the corresponding averages are 6.6, 0.7, and 7.5. These results demonstrate that we observe the same phenomenon across multiple datasets and multiple features, which verifies that the finding is robust rather than anecdotal.
> ***
> [P&M2] In our setting, the test data is unseen during training, so it is more similar to domain generalization. Domain adaptation is also an important aspect, which we will include in the related work section.
> ***
> [P&M3] We agree with your observation that imputing missing features effectively changes the data distribution. In our problem, we define the combination of the two as a *coupled shift*—addressing either shift individually cannot adequately solve the problem. What we want to emphasize is that our method works synergistically as a whole: it analyzes and addresses both components of the coupled shift, achieving strong performance.
> ***
> [P&M4]
> First, entire-column missingness occurs in many real-world scenarios, such as sensor failures in industrial monitoring or instrument unavailability in medical testing. Regarding the case of partial missingness that you mentioned, we have also conducted experiments and found that our method still provides improvements under partial missingness.
>
> | Model                          | 0.2    | 0.4    | 0.6    | 0.8    |
> |--------------------------------|--------|--------|--------|--------|
> | (ANES)Light GBM                | 79.89  | 79.87  | 79.89  | 79.88  |
> | (ANES)Light GBM + ours         | 80.30  | 80.31  | 80.30  | 80.28  |
> | (Hypertension)LightGBM         | 57.25  | 57.24  | 57.26  | 57.27  |
> | (Hypertension)LightGBM + ours  | 63.88  | 63.89  | 63.88  | 63.92  |
>
> At the same time, we would like to clarify that for partial missingness, one could design methods to leverage the remaining information. Our method, without using such information, can still demonstrate strong performance, highlighting its robustness and practical value.
>
> ***
> [Overall] Based on the paper and our responses, our observations on the RTCS problem represent a widespread phenomenon rather than a peculiarity of a single dataset. This highlights the significance of our observations and the rationale for designing solutions based on them. Guided by your valuable comments, the revised version of our observations would make the problem formulation more compelling.

---

> ### Author Response · Authors · 2025-11-15
> **Method and Experiments section**
>
> We sincerely appreciate your valuable feedback on our work. We address each of your concerns in detail below.
> ***
> ## Method section
> ***
> [M1] During the process of handling each column independently, the imputation rules proposed by the LLM make use of other features in the table. Therefore, the correlations between table features are carefully taken into account in the imputation process.
> ***
> [M2] LightGBM is faster than CatBoost. In the process of validating rules, we do not want to spend too much time.
> ***
> [M3] In the experimental setup, we mentioned that we use DeepSeek-V3 as the rule proposer, and in Appendix G we present the prompts and model responses used. Regarding your question about different model architectures, we conducted experiments with Qwen-3, and the results are as follows:
>
>
> | Model               | ANES    | Hypertension |
> |---------------------|---------|--------------|
> | CatBoost            | 79.86   | 58.32        |
> | (Qwen3-max) Ours+CatBoost         | **80.41**   | **62.58**        |           |
> | LightGBM            | 79.87   | 56.47        |
> | (Qwen3-max) Ours+LightGBM         | **80.29**   | **62.99**        |
>
> The results indicate that our designed framework is compatible with multiple language model architectures.
> ***
> ## Experiment section
> ***
> [E1&E2] In existing studies, testing for missing values is mostly conducted by manually injecting masks. Therefore, we followed their approach. Regarding the method you suggested for validation on real data, we have supplemented the results on TableFSBench. The results also include outcomes for both multi-class and regression tasks, demonstrating the effectiveness of our method.
>
>
> | Model           | eyemovements (multi-class)-ACC | jannis (multi-class)-ACC | concrete(regression)-RMSE |
> |-----------------|--------------------------------|--------------------------|---------------------------|
> | CatBoost        | 64.35                          | 69.04                    | 24.95                     |
> | LightGBM        | 64.21                          | 68.48                    | 21.38                     |
> | ERM             | 42.18                          | 58.95                    | 15.93                     |
> | IRM             | 44.74                          | 59.56                    | 38.59                     |
> | vREX            | 45.15                          | 59.35                    | 15.88                     |
> | EQRM            | 42.46                          | 61.13                    | 15.89                     |
> | CatBoost+Ours   | **65.21**                          | **70.95**                    | **12.30**                     |
> | LightGBM+Ours   | **68.00**                          | **70.61**                    | **12.73**                     |
>
> ***
> [E3] An explanation for the relatively smaller performance improvement observed on the ANES dataset is that the discriminative power of features with respect to the label affects the performance gain. To demonstrate this, we used information gain to measure the discriminative power of features in the datasets. We calculated the information gain for the training and test sets (with distribution shift and missing features) of both the ANES and Hypertension datasets. The results show that the maximum information gain for the test set of ANES decreased by 0.012 (from 0.079 to 0.067), whereas for Hypertension, it increased by 0.027 (from 0.033 to 0.060). Therefore, the lower discriminative power of features with respect to the label within the ANES dataset indirectly led to a lower performance gain.

---

> > ### Author Response · Authors · 2025-11-15
> > **Method and Experiments section (Contd.)**
> >
> > ***
> > [E4] For the imputation module, our method focuses on proposing an interpretable and semantically rich imputation rule to fill in missing values while ensuring robustness. Compared with training a dedicated imputation model, this approach is simpler and does not aim to minimize the difference between the imputed values and the original values. For comparisons with imputation baselines, we adopt a fill-train-infer paradigm. Since baseline methods such as k-NN are ineffective when an entire column is missing, we use MICE as an example, and LinearRegression as baseline imputation methods. We show the performance of combining these baseline imputation methods with our approach. It is worth noting that for the RTCS problem, training a dedicated imputation model does not solve the problem effectively. As [P&M3] points out, RTCS is a coupled variation problem, which must be addressed as a whole; solving it individually does not yield good performance.
> >
> > | Model          | ANES    | Hypertension | ACSincome | ACSpubco |
> > |----------------|---------|--------------|-----------|----------|
> > | LightGBM       | 79.87   | 56.48        | 76.74     | 64.68    |
> > | MICE           | 79.85   | 59.73        | 79.36     | 61.61    |
> > | LR             | 79.78   | 59.57        | 80.04     | 65.03    |
> > | LightGBM+Ours  | **80.38**   | **63.75**        | **80.86**     | **73.59**    |
> >
> > ***
> > [E5] Our test data remains unseen during training, so there is no data leakage. It should be noted that when the missing columns are unknown, we can prompt the construction of imputation rules for each column and train using our method. This is feasible in practical deployment, as our approach offers advantages in both storage and computation time (Appendix E). Since detecting entirely missing columns is relatively straightforward—requiring only counting the number of NaNs in each column—our method can still be effective even when the missing columns are unknown.
> > ***
> > Ref:
> >
> > [1] TabFSBench: TabFSBench: Tabular Benchmark for Feature Shifts in Open Environments. In: ICML 2025
> >
> > [2] MICE: Multiple imputation using chained equations: issues and guidance for practice. In: Statistics in medicine

---

> ### Author Response · Authors · 2025-11-15
> **Questions section**
>
> We sincerely appreciate your valuable feedback on our work. We address each of your concerns in detail below.
> ***
> [Q1] Not only IRM, but other methods also exhibit similar results with Table 1, as shown in the table below. All the shown performance is lower than a TabM model.
>
> | Model          | ANES       | Hypertension | ACSincome   | ACSpubcov   |
> |----------------|------------|--------------|-------------|-------------|
> | ERM            | 79.13      | 56.87        | 77.91       | 48.10       |
> | ERM+Masked     | 79.43      | 57.30        | 78.27       | 47.31       |
> | EQRM           | 78.22      | 56.40        | 77.83       | 47.48       |
> | EQRM+Masked    | 78.47      | 54.59        | 78.24       | 49.75       |
>
> ***
>
> [Q2] We appreciate the reviewer’s concern regarding the clarity of the Knowledge-Guided Feature Aligner and the imputation rule. To address this, Appendix G provides the full prompt used for rule generation, detailed examples of the resulting rules, and the corresponding Python implementation of the proposed rule-based imputation procedure.
> ***
> [Q3] Since the rules proposed by the LLM mainly rely on other features in the table, the resulting imputation rules may differ from the original semantics. Our goal is for the LLM to leverage the existing features to propose a practically meaningful imputation rule, so the semantic meaning of the feature combinations used will be similar to—but not identical with—the true meaning of the column. Therefore, we refer to this as a semantic gap. Under this condition, the model does not have sufficient learning on the newly imputed data, so directly applying the rule and then predicting will not yield good performance.

---

### Official Review · Reviewer_48J6 · 2025-10-30

**Soundness:** 3
**Presentation:** 2
**Contribution:** 3
**Rating:** 4
**Confidence:** 4

**Summary:**

This paper investigates the Robust Tabular Prediction under Coupled Shifts (RTCS) problem, where feature missingness and distributional shifts jointly affect performance. The authors propose KGCS4Tab, a knowledge-guided framework that disentangles missingness from distribution shifts through recovery rules and adaptive model selection. Experiments show that KGCS4Tab achieves promising performance improvement.

**Strengths:**

S1: The motivations are clearly and thoroughly analyzed.

S2: The applicability of this specific imputation setting is validated through real-world scenarios.

**Weaknesses:**

W1: The theoretical analysis section (Theorem analysis) focuses on the probabilistic selection but is loosely connected to the imputation task. If the authors intend to devote such a large portion to theoretical content, they should provide a more explicit justification of how the theoretical results relate to or benefit the imputation process.

W2: The mechanism of the Proposed Rule remains unclear. It seems capable of directly predicting missing values, raising the question of why an LLM–based imputation approach, possibly through generated code, would not suffice. The paper should clarify why the proposed complex pipeline is necessary and how it outperforms a simpler LLM-based alternative.

W3: Line 195 mentions a distribution shift across domains, yet the experiments appear to be conducted within the same domain. The paper would be stronger with explicit cross-domain evaluations to support this claim.

W4: The presentation quality needs improvement. For example, line 179 contains redundant sentences. Given some example of the generated rules could enhance the presentation of the paper.

**Questions:**

Nan

---

> ### Author Response · Authors · 2025-11-15
>
> We sincerely appreciate your valuable feedback on our work. We address each of your concerns in detail below.
> ***
> [W1]
> We appreciate this comment and clarify the direct connection between our theoretical results and the imputation task.
> The first theorem ensures that computations remain scale-consistent across different feature types, while the second theorem provides an upper bound on the error introduced by the probabilistic selection strategy. In our method, the training set is augmented through imputation during training, and therefore contains information about the imputation rules. As a result, Theorem 1 guarantees the consistency of distributional computations between the imputation-augmented training data and the test data, which is essential for accurately measuring distribution distances across imputed columns. Since the training set encodes the imputation rules, the term ($\Delta$) in Theorem 2 also reflects the influence of the imputation process. Consequently, the derived error bound naturally incorporates the additional error introduced by imputation. This bound demonstrates the robustness of our method to the imputation procedure while also ensuring computational efficiency. We will clarify these points in the revised version.
> ***
> [W2] The proposed rules are generated by combining the background information of the dataset with the knowledge of a large language model (LLM). However, because the rules produced by the LLM primarily rely on other features within the table, the inferred recovery rules may deviate from the original underlying semantics. As a result, directly applying LLM-based imputation yields suboptimal performance, as shown in the table below. Our method effectively bridges this semantic gap through training, making it substantially more effective.
>
> | Method          | ANES  | Hypertension |
> |-----------------|-------|--------------|
> | LightGBM        | 79.87 | 56.48        |
> | LightGBM+direct_imputation | 79.80 | 56.68        |
> | LightGBM+Ours   | **80.38** | **63.75**        |
>
>
> ***
> [W3] The experimental results reported in line 195 are obtained under a cross-domain setting. Specifically, the model is trained on the source domain and evaluated on the target domain. We will clarify this setup in the revision.
> ***
> [W4] Thank you for the suggestion. We have removed it in the revision, and the examples of the generated rules are provided in Appendix G.

---

### Author Response · Authors · 2025-11-27
**Overall Reply**

We are very grateful to the reviewers for their valuable feedback. We have revised the paper and uploaded the updated version for your review and discussion. The modifications are summarised as follows:
***
**[Related Works]**

In the Related Works section, we thank Reviewer Q3ez (PM2) and Reviewer 6gUQ (W1) for their suggestions. We have added discussions on the domain adaptation field and on LITO, an advanced method for solving table-related problems using LLMs.

**[3.2 Problem Analysis]**

* **Observation 1:** We thank Reviewer 48J6 (W4) for the suggestion; we have removed redundant sentences. We also thank Reviewer Q3ez (PM1) for pointing out the performance drop in the datasets, and we have provided a deeper analysis (C.4) to demonstrate that our observation is a general phenomenon.
* **Observation 3:** We thank Reviewer 48J6 (W3) for clarifying the experimental conditions in Fig. 4 (line 195).

**[4.1 Knowledge-Guided Feature Aligner]**
We thank Reviewer Q3ez (Q3) for the comment; we have further clarified the meaning of the semantic gap.

**[4.3 Theoretical Analysis]**
We thank Reviewer 48J6 (W1); we have clarified the relationship between our theoretical analysis and the imputation process.

**[5.1 Empirical Results RQ2]**
We thank Reviewers Q3ez (E4), 6gUQ (W1), and 48J6 (W2); we have added comparison results with the direct rule-based approach and the impute-then-train baseline. The experiments demonstrate that our method outperforms the two aforementioned paradigms in RTCS tasks.

**[5.1 Empirical Results RQ3]**
We thank Reviewer Q3ez (PM4); we have supplemented performance results under different missing rates when feature parts are missing. Our method still maintains good performance in this setting.

**[5.2 Further Analysis – Effect of Number of Source Models]**
We thank Reviewer nxRD (Q2); we have added results compared with direct ensemble in C.7. The experiments demonstrate the superiority of our selection strategy.

**[5.2 Further Analysis – Adaptation to Different Model Structures]**
We thank Reviewer Q3ez (M3); we conducted experiments by replacing the rule proposer with Qwen-series models. The experimental performance remains robust, demonstrating the robustness of our method to LLM architectures.

**[5.2 Further Analysis – Performance on Complex Tasks]**
We thank Reviewer Q3ez (E1 & E2); we have added experiments on three larger datasets in TableFSBench, including multi-classification and regression tasks. The results show the effectiveness of our method.

**[D.2 Rule Correlations and Performance Recovery Ratios]**
We thank Reviewer 6gUQ (W2); we analysed rule correlations and the upper-bound ratios of performance recovery.

**[E Time and Memory Cost]**
We thank Reviewers nxRD (W2) and Q3ez (E5); we have added a discussion on performance cost when missing columns are unknown.
***
Regarding other questions, we have provided corresponding responses in the paper and in our replies:

**[Q3ez PM3, PM-Overall; nxRD W1]**

Regarding the research problem, we study the scenario where feature missingness and distribution shifts are coupled. Addressing only one type of change cannot adequately solve this problem. We emphasise that our method works as an integrated system, simultaneously analysing and handling the two components of coupled shifts. This is a unique challenge. Therefore, this is not a simple A+B scenario: the combination of multiple challenges introduces new difficulties that do not arise in a mere A+B setting.

**[Q3ez M1]**  In the process of handling each column independently, the imputation rules proposed by the LLM leverage other features in the table. Thus, feature correlations are fully considered. The experimental results in D.2 also demonstrate this point.

**[Q3ez M2]** LightGBM runs relatively fast. In our validation environment, we aimed to avoid unnecessary time cost. This model balances efficiency and performance.

**[Q3ez E3]** Regarding the relatively small performance improvement on the ANES dataset, we conducted a quantitative analysis from the perspective of information gain (lines 419–428).

**[Q3ez Q1]** The experimental results indicate that the issue occurs not only in IRM but also in other methods. Due to space limitations, we did not modify Table 1.

**[Q3ez Q2]** The generated rules are presented in detail in Appendix G, including the rationale behind each rule and the corresponding executable Python code.

**[nxRD Q1]** We chose to compare with LLM-based methods because they leverage the background knowledge of LLMs, which may provide better robustness to coupled shifts. The experimental results, where TabLLM achieves relatively strong performance on some datasets, also support this point. Under the same settings, CatBoost runs faster than our method, as our approach introduces additional distance computations.
***
We sincerely thank all the reviewers for their insightful and constructive comments, which have greatly helped us improve the paper.

---

### Author Response · Authors · 2025-12-02

Dear AC, SAC, and PC,
Considering the reassignment of the AC, I would like to provide a summary of the reviewers’ comments and our responses. We greatly appreciate the valuable feedback from all reviewers and your work.
***
**48J6**

**W1** Regarding the relationship between the theoretical analysis and the imputation process, our theory is directly connected to the imputation procedure. We have added an explanation in *Remark 3* of the revised manuscript.

**W2** In response to the concern that *“The mechanism of the proposed rule remains unclear,”* we conducted experiments showing that directly applying the rule does not achieve good performance, and that our method utilizes the rule more effectively. We revised the discussion of this result in *RQ2* of the paper.

**W3 & W4** Concerning the experimental setup around line 195, we added further explanation in Section *3.2 Problem Analysis* of the revised version and removed redundant sentences. Examples of the generated rules and case studies are provided in *Appendix G*.
***

**Q3ez**

**PM1:** Regarding whether the phenomenon illustrated in Fig. 2 is universal, we conducted a deeper experimental analysis to demonstrate that our findings are indeed general. This includes results on additional datasets as well as experiments involving missing values in other columns of the same dataset. We added this analysis in Section *3.2 Problem Analysis* of the revised manuscript.

**PM2:** We added a discussion of domain adaptation to the *Related Works* section and revised the relevant parts accordingly.

**PM3:** Concerning the conceptual relationship between the two types of shifts, we agree with the reviewer that once a missing feature is imputed, the underlying data distribution changes. Therefore, we define this as a **coupled shift** in the paper. Addressing only one type of shift is insufficient for solving the problem effectively, which highlights the necessity and effectiveness of our proposed method.

**PM4:** Regarding model performance under partially missing columns, we conducted additional experiments. The results, revised in *5.1 Empirical Results (RQ3)*, show that our method remains robust even when columns are only partially missing.

**M1:** For the concern about feature correlation during column recovery, although we process each column independently, the imputation rules generated by the LLM **leverage other features in the table**, ensuring that inter-feature correlations are well captured during the recovery process.

**M2:** Regarding the choice of validation models, we selected LightGBM primarily due to its training efficiency.

**M3:** To address concerns about the method’s adaptability to different LLM architectures, we replaced the backbone model with the Qwen series. The performance remained consistently strong. The revised results are reported in *5.2 Further Analysis – Adaptation to Different Model Structures*.

**E1 & E2:** Regarding dataset and task selection, we followed standard practice in existing literature, where missing values are typically introduced synthetically for evaluation. Additionally, we included multi-class classification and regression tasks to further validate the effectiveness of our method. The revised results appear in *5.2 Further Analysis – Performance on Complex Tasks*.

**E3:** For the relatively small improvement on the ANES dataset, we explained the reason from an information-entropy perspective in the original manuscript (lines 419–428).

**E4:** In response to the request to compare our method with traditional impute–train–predict pipelines, we conducted additional experiments using several widely known models. The results demonstrate that our method achieves superior performance. The updated results appear in *5.1 Empirical Results (RQ2)*.

**E5:** Regarding concerns about our experimental setup, we clarify that the test data remained unseen throughout training, so there is no data leakage. Moreover, when the missing column is unknown, we can prompt the model to generate imputation rules for each column and train accordingly. This is feasible in deployment, as our approach is efficient in both storage and computation (see Appendix E). Additional clarifications are provided in *E Time and Memory Cost* of the revised version.

**Q1:** Concerning the generality of Table 1 results, we conducted the same experiments using other methods and observed consistent phenomena. Due to space limitations, Table 1 itself was not modified.

**Q2:** Regarding rule generation, the revised *Appendix G* provides detailed examples of the generated rules, including the rationale behind each rule and the corresponding executable Python code.

**Q3:** Regarding the semantic gap, we further clarified its significance in Section *4.1 Knowledge-Guided Feature Aligner* of the revised manuscript.

---

> ### Author Response · Authors · 2025-12-02
> **(Contd.)**
>
> **6gUQ**
>
> **W1:** Regarding the comparison with LLM-based imputation methods, we first included additional discussion of the existing LITO work in the revised manuscript. LITO mainly addresses class-imbalance issues, which differ substantially from the focus of our study. For a more comprehensive comparison, we conducted experiments comparing our method with traditional *imputation + model inference/training* pipelines. These results have been revised in *5.1 Empirical Results (RQ2)*. Additionally, we performed a coarse comparison with ATLLM and briefly presented the results in our response.
>
> **W2:** Concerning the question on predefined correlation rules, we conducted experiments using feature importance from tree-based models. The results show that our method can capture semantic relationships between columns in both clear and subtle scenarios, and the recovered rules generally include the relevant columns.
> Regarding the question of the *upper bound for performance gains*, we compared our method with the results obtained using GT (ground-truth label distribution). The findings indicate that our method approaches the upper bound of achievable performance improvements.
> Revisions addressing both issues have been added to *D.2 Rule Correlations and Performance Recovery Ratios*.
>
> ---
>
> **nxRD**
>
> **W1:** To clarify the RTCS issue: our paper focuses on the **coupled shift** caused jointly by distribution shift and feature missingness. As shown in our analysis, addressing only distribution shift or only feature missingness cannot solve this problem. Therefore, this is not a simple *A + B* scenario; the combination of multiple factors leads to new challenges that do not arise in standard additive problems.
>
> **W2:** Because our method incurs only small storage and computational overhead (Appendix E), it remains practical even when many columns are potentially missing. Moreover, we require only **two LLM API calls** to generate a new rule, so the computational cost is minimal.
>
> **Q1:** Regarding computation speed, our method meets practical time-efficiency requirements. It introduces additional distance computations compared with the base model, so the runtime increases slightly, but remains acceptable.
>
> **Q2:** Concerning the comparison with direct ensembling: we performed experiments comparing directly ensembling the trained models with our method. The results show that our approach significantly outperforms direct ensembling, **highlighting the effectiveness of our method**. Revisions appear in *5.2 Further Analysis – Effect of Number of Source Models*.
>
> **Note:** As the only reviewer who provided feedback during the discussion stage, the nxRD reviewer noted that our previous responses successfully addressed their questions. They raised one additional concern regarding the computational cost of LLM calls. We clarified that **only two LLM API calls** are needed to generate each new rule, resulting in negligible overhead.
> ***
> We sincerely thank the AC, SAC, and PC for their efforts, and we are also grateful to all reviewers for their valuable comments.

---

### Meta-Review · Area_Chair_Xm7a · 2025-12-14

**Summary:**

The reviewers raise the following major concerns:

1) The theoretical analysis does not sufficiently inform or justify the practical implementation (48J6).

2) The motivation and reasoning behind key design choices are not adequately explained (48J6).

3) The method lacks validation across diverse domains (48J6).

4) A more rigorous conceptual and theoretical distinction between feature missingness and distribution shift is required (Q3ez).

5) The assumption that entire columns are missing does not reflect real-world scenarios where partial feature missingness is more common (Q3ez).

6) The paper does not discuss why LightGBM was chosen over potentially stronger alternatives like CatBoost (Q3ez).

7) The impact of different LLM choices on method performance and robustness is not adequately addressed (Q3ez).

8) Experiments lack critical components, including naturally-occurring missingness, realistic scenarios without prior knowledge of missing columns, and comparisons with LLM-based imputation methods (Q3ez, 6gUQ).

9) The paper does not discuss the theoretical or empirical upper bounds of the proposed method (6gUQ).

10) The specific problem formulation is one of many possible framings (nxRD).

11) The method lacks computational scalability (nxRD).

**Reviewer Concerns:**

Concerns addressed by rebuttal: 2); 6); 7); 8); 9) The rebuttal partially answers the question, but doesn't provide a full explanation for the observation.

Outstanding concerns: 1) It's still unclear how the theoretical analysis is connected to the implementation; 3) The cross-domain evaluation is limited; 4); 5) The partial missingness is not fully addressed (e.g., what's the underlying missing mechanism); 10); 11)

**Reviewer Scores:**

Each reviewer's comments are partially addressed, but there are still outstanding concerns. They would keep or slightly increase their scores.

---

### Decision · Program_Chairs · 2026-01-26

Reject